# Four-dimensional Variational Assimilation for $SO_2$ Emission and its Application around the COVID-19 lockdown in the spring 2020 over China

Yiwen Hu[1,2], Zengliang Zang[2], Xiaoyan Ma[1], Yi Li[2], Yanfei Liang[3], Wei You[2], Xiaobin Pan[2], Zhijin Li[4,5]

[1]Key Laboratory for Aerosol-Cloud-Precipitation of China Meteorological Administration, Nanjing University of Information Science & Technology, Nanjing, 210044, China
[2]College of Meteorology and Oceanography, National University of Defense Technology, Changsha, 410073, China
[3]No. 32145 Unit of PLA, Xinxiang, 453000, China
[4]Department of atmospheric and oceanic sciences, Fudan University, Shanghai, 200031, China
[5]University of California Los Angeles, California, 91109, USA

*Correspondence to*: Zengliang Zang (zzlqxxy@163.com) and Xiaoyan Ma (xma@nuist.edu.cn)

**Abstract.** Emission inventories are essential for modeling studies and pollution control, but traditional emission inventories are usually updated after a few years based on the statistics of "bottom-up" approach from the energy consumption in provinces, cities, and counties. The latest emission inventories of Multi-Resolution Emission Inventory in China (MEIC) was compiled from the statistics for the year 2016 (MEIC_2016). However, the real emissions have varied yearly, due to national pollution control policies and accidental special events, such as the coronavirus disease (COVID-19) pandemic. In this study, a four-dimensional variational assimilation (4DVAR) system based on the "top-down" approach was developed to optimize sulfur dioxide ($SO_2$) emissions by assimilating the data of $SO_2$ concentrations from surface observational stations. The 4DVAR system was then applied to obtain the $SO_2$ emissions during the early period of COVID-19 pandemic (from 17 January to 7 February, 2020), and the same period in 2019 over China. The results showed that the average MEIC_2016, 2019, and 2020 emissions were $42.2 \times 10^6$, $40.1 \times 10^6$, and $36.4 \times 10^6$ kg d$^{-1}$. The emissions in 2020 decreased by 9.2% in relation to the COVID-19 lockdown compared with those in 2019. For Central China, where the lockdown measures were quite strict, the mean 2020 emission decreased by 21.0% compared with 2019 emissions. Three forecast experiments were conducted using the emissions of MEIC_2016, 2019, and 2020 to demonstrate the effects of optimized emissions. The root-mean-square error (RMSE) in the experiments using 2019 and 2020 emissions decreased by 28.1% and 50.7%, and the correlation coefficient increased by 89.5% and 205.9% compared with the experiment using MEIC_2016. For Central China, the average RMSE in the experiments with 2019 and 2020 emissions decreased by 48.8% and 77.0%, and the average correlation coefficient increased by 44.3% and 238.7%, compared with the experiment using MEIC_2016 emissions. The results demonstrated that the 4DVAR system effectively optimized emissions to describe the actual changes in $SO_2$ emissions related to the COVID lockdown, and it can thus be used to improve the accuracy of forecasts.

# 1 Introduction

Sulfur dioxide ($SO_2$) causes acid rain through the formation of sulfuric acid, which destroys infrastructure and harms aquatic and terrestrial ecosystems (Saikawa et al., 2017; Zheng et al., 2018). $SO_2$ is also a precursor of sulfate aerosols, which directly affect the radiation budget and indirectly modulate clouds and precipitation, and also cause haze pollution (Qin et al., 2022). Thus, $SO_2$ emission impacts the ecological environment. $SO_2$ pathway in the atmosphere is generally investigated using chemistry transport models (CTMs) to estimate the three-dimensional changes of $SO_2$ concentrations. Thus, accurately estimating $SO_2$ emissions is important for understanding spatiotemporal distribution of $SO_2$ concentrations in CTMs (Zeng and Wu, 2021).

$SO_2$ emissions are generally estimated using the "bottom-up" approach, which requires direct observations of the activities and emissions factors from all possible sources (Zhao et al., 2022). However, the estimates are subject to substantial uncertainties because of limited available observations, with the differences among existing inventories as high as 42% (Granier et al., 2011). Saikawa et al. (2017) compared five types of emission inventories and found a significant difference in $SO_2$ emissions from power sector due to the difference in the assumed installation period of flue gas desulfurization in coal-fired power plants. Moreover, most "bottom-up" emissions are recorded annually or monthly amounts, which need to be spatiotemporally allocated into the hourly gridded emissions for use in regional air quality models, and thus can cause uncertainties (Peng et al., 2017; Peng et al., 2018; Zeng and Wu, 2018). China has implemented several control strategies, such as strengthening emission standards, phasing out obsolete industrial capacity, and establishing small-but high-emitting factories (Zheng et al., 2018), all of these have markedly reduced the emissions. However, these policies have been applied to varying extents in different regions, so that emission changes vary spatiotemporally (Chen et al., 2019a; Dai et al., 2021). Such complex changes in $SO_2$ emission were not reflected in the "bottom-up" estimates. Differences in the spatiotemporal control also caused additional uncertainties in gridded hourly emissions reducing their accuracy (Zeng et al., 2020).

In contrast to the "bottom-up" approach, data assimilation (DA) provides a "top-down" approach, where the ensemble Kalman filter (EnKF) and four-dimensional variational DA (4DVAR) are two of the most explored algorithms to optimize emissions (Cohen and Wang, 2014; Wang et al., 2021). The EnKF method uses flow-dependent covariance generated by an ensemble of model outputs to convert observational information into emissions (Tang et al., 2013; Ma et al., 2019), and has been used to estimate regional and global aerosols and gas-phase emissions, such as $SO_2$, NOx, CO, and particulate matter. (Huneeus et al., 2012; Huneeus et al., 2013; Miyazaki et al., 2012; Miyazaki et al., 2014; Tang et al., 2013; Tang et al., 2016; Chu et al., 2018). For example, Dai et al. (2021) developed a four-dimensional regional ensemble transform Kalman filter and showed that $SO_2$ emissions over China in November 2016 decreased by 49.4% in comparison to the 2010 background emission due to the implementation of emission control policies (Zheng et al., 2018). Peng et al. (2017, 2018) developed an EnKF system to include more spatiotemporal emission characteristics over China using hourly surface observations as constraints, and the forecasting results with optimized emissions are more accurate than those with the background emissions.

The $SO_2$ forecasts with the optimized emissions were improved for the forecast out to 72-h, and the root-mean-square errors (RMSEs) decreased by 30% in comparison to the forecasts with the background emission. Feng et al. (2020) quantitatively optimized the gridded CO emissions in China using hourly surface CO measurements and EnKF algorithm with the Weather Research and Forecasting (WRF)/CMAQ model, and found the optimized CO emissions in December 2017 17% lower than those in December 2013.

A 4DVAR method has been used to estimate emissions based on the adjoint model of a CTM and is known as an inverse process (Bao et al., 2019; Yumimoto and Uno, 2006; Yumimoto et al., 2007; Wang et al., 2021). Several studies have shown that 4DVAR is a promising approach to derive the emission rates (Dubovik et al., 2008; Hakami et al., 2005; Müller and Stavrakou, 2005; Elbern et al., 2007; Yumimoto et al., 2007; Yumimoto et al., 2008). Stavrakou and Atmospheres (2006) estimated CO and NOx emissions with a 4DVAR system using satellite data as a constraint and showed

that the optimized CO emission was 2900 Tg $yr^{-1}$, which was about 5% higher than the background emission. Henze et al. (2007) developed an adjoint model based on the GEOS-Chem model and used it to optimize the SOx, NOx, and $NH_3$ emissions. The model was also used to investigate the sensitivity of modeled aerosol concentrations to their precursor emissions, suggesting that their relationship strongly depended on thermodynamic competition. Qu et al. (2019) estimated $SO_2$ emissions by assimilating OMI observations using the GEOS-Chem model and its adjoint model and found that the $SO_2$

emissions decreased by 48% over China from 2008 to 2016. The emissions based on a "top-down" approach can reduce the uncertainty of "bottom-up" emissions and provide a more accurate emission related to a special event than traditional emissions.

Emergence of the coronavirus disease (COVID-19) pandemic during the period from the end of 2019 through the beginning of 2020 (Wang et al., 2020) impacted more than 200 countries. To slow and stop the rapid spread of the virus,

Wuhan was the first city to implement a lockdown on January 23, 2020, followed by the entire Hubei province one day later (Wuhan is capital of the Hubei province). Subsequently, all provinces in China successively implemented a national emergency to respond to major public health emergencies. The pollutant emissions decreased because human activities reduced during the lockdown (Filonchyk et al., 2020; Forster et al., 2020; Ghahremanloo et al., 2021; Keller et al., 2021; Li et al., 2020; Miyazaki et al., 2020; Li et al., 2021; Huang et al., 2021a; Zhang et al., 2020). For example, Huang et al. (2021b)

estimated NOx emissions over China during this period and found a decrease trend owing to human activity reduction.

In this study, we developed a 4DVAR system to estimate $SO_2$ emissions, using the WRF model coupled with chemistry (WRF-Chem) (Grell et al., 2005). Some physical and chemical processes, including transport, dry/wet deposition, emission, vertical mixing, and $SO_2$ chemicals, were implemented to describe the pathway of $SO_2$ in WRF-Chem. The 4DVAR system was applied to investigate the changes in $SO_2$ emissions over China during the COVID-19 lockdown. Hourly surface $SO_2$

observations were assimilated.

This paper is organized as follows. Section 2 describes the methodology, including the WRF-Chem and 4DVAR system configurations and their adjoint model, as well as observational data. In Section 3, the spatiotemporal changes in $SO_2$

emission during the COVID-19 lockdown are estimated. SO₂ simulations using optimized emissions are also verified against observations to show the improvements in emission data. Finally, a discussion and conclusions are presented in Section 4.

## 2 Method and Data

### 2.1 WRF-Chem model

WRF-Chem is an online coupled air quality model (Grell et al., 2005), which includes sophisticated and comprehensive physical and chemical processes such as transport, turbulence, emission, chemical transformation, photolysis, radiation, and more. The WRF-Chem version 3.9.1 was used in this study. The WRF-Chem domain (Fig. 1a) is centered at 101.5 °E, 37.5 °N, and covers all of China with 27 km horizontal resolution. There are totally 169×211 grid points. In the vertical, 40 vertical layers extend from the surface to 50 hPa, with high resolution near the surface. Meteorological initial and boundary conditions were derived from the 1° × 1° National Centers for Environmental Prediction Global Final Analysis data at a 6-h frequency. Most of the WRF-Chem settings follows Hu et al. (2022) (Table 1). Those settings include the WRF Lin microphysics scheme (Lin et al., 1983), Rapid Radiative Transfer Model longwave (Mlawer et al., 1997), Goddard shortwave radiation schemes (Chou, 1994), Yonsei University (YSU) boundary layer scheme (Hong et al., 2006), Noah land surface model (Chen et al., 2010), and Grell-3D cumulus parameterization (Grell, 1993; Grell and Dévényi, 2002). Aerosol and gas-phase chemistry schemes are the aerosol interactions and chemistry (MOSAIC-4 bin) and carbon bond mechanism-Z (CBMZ) (Zaveri and Peters, 1999; Zaveri et al., 2008). The heterogeneous SO₂ reaction is also added to the WRF-Chem (Sha et al., 2019). The anthropogenic emissions from the Multi-Resolution Emission Inventory for China (MEIC) in 2016 are used as the background emission input.

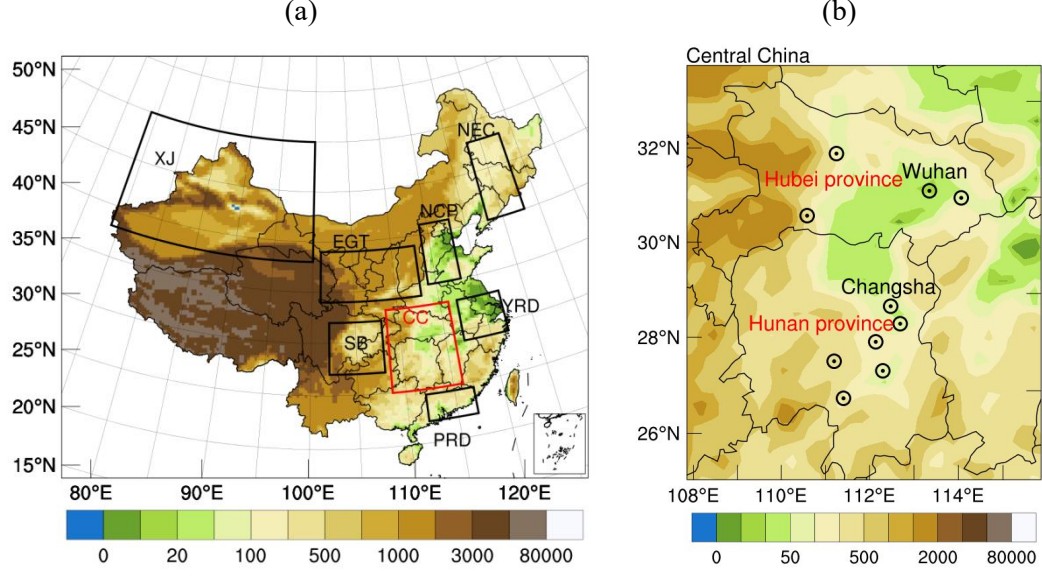

(a)                                          (b)

Figure 1: (a) Maps of the WRF modeling domain and (b) Central China. The color bars represent the terrain altitude. The black rectangles in (a) are North China Plain (NCP), Northeastern China (NEC), Energy Golden Triangle (EGT), Xinjiang (XJ), Sichuan Basin (SB), Yangtze River Delta (YRD), and Pearl River Delta (PRD). The red rectangle in (a) represents Central China (CC). (b) The details of CC. Black circle with dots in (b) represent the locations of large cities. The red characters in (b) are the name of provinces, and the black characters are the name of capital cities. Wuhan and Changsha are the capitals of Hubei and Hunan provinces. Inset in (a): South China Sea. Units: m.

Table 1: WRF-Chem model configuration.

| Physical or chemical process | Option |
| --- | --- |
| Microphysics | Lin microphysics scheme (Lin et al., 1983) |
| Longwave radiation | Rapid Radiative Transfer Model longwave (Mlawer et al., 1997) |
| Shortwave radiation | Goddard Space Flight Center shortwave radiation scheme (Chou, 1994) |
| Boundary layer scheme | Yonsei University (Hong et al., 2006) |
| Land surface model | Noah land surface model (Chen et al., 2010) |
| Cumulus parameterization | Grell 3-D scheme (Grell, 1993; Grell and Dévényi, 2002) |
| Aerosol scheme | Model for Simulating Aerosol Interactions and Chemistry (MOSAIC-4 bin) (Zaveri et al., 2008) |
| Gas scheme | Carbon Bond Mechanism-Z (CBMZ) (Zaveri and Peters, 1999) |

## 2.2 4DVAR system

4DVAR is a continuous data assimilation method to simultaneously assimilate a time series of observations over a time window. It produces an analysis that fit a set of observations taken over the time window. The time evolution of the concerning quantities is governed using a deterministic model as a strong constraint. The cost function of 4DVAR can be written as follows:

$$J = \frac{1}{2}\left(c_0 - c_0^b\right)^T \mathbf{B}_c^{-1}\left(c_0 - c_0^b\right) + \frac{1}{2}\sum_{i=0}^{n-1}\left(e_i - e_i^b\right)^T \mathbf{B}_{ei}^{-1}\left(e_i - e_i^b\right) + \frac{1}{2}\sum_{i=0}^{n}\left(y_i^o - \mathbf{H}_i c_i\right)^T \mathbf{R}_i^{-1}\left(y_i^o - \mathbf{H}_i c_i\right) \tag{1}$$

where $c_0$ and $e_i$ are the control variables to denote the initial concentration vector and the emission to estimate. $c_0^b$ is the background concentration at zero time, and $e_i^b$ is the background emission. The subscripts of the variables represent time levels, and $n$ is the time window. The first term in Eq. (1) is the background term due to the initial concentration field, and the second term is the background term for emissions in the time window. $\mathbf{B}_c$ and $\mathbf{B}_{e_i}$ are the background error covariances (BEC) for the initial concentrations and background emissions. The third term is the observation term, where

$y_i^o$ is the observation vector at time $i$. $\boldsymbol{H_i}$ is the observation operator maps the control variables to the observations, and $\mathbf{R_i}$ is the observation error covariance matrix. The concentration $c_i$ is governed by a model.

$$c_i = f_{i,i-1}(c_{i-1}, e_{i-1}) \tag{2}$$

where $f_{i,i-1}$ represents the model time integration for one time step from time $i-1$ to $i$. The increment field of the initial SO$_2$ concentration can be written as $\delta c_0 = c_0 - c_0^b$, and the increment field of SO$_2$ emission as $\delta e_i = e_i - e_i^b$. The innovation vector is denoted as $d_i \equiv y_i^o - \boldsymbol{H_i}(c_i)$, which is the difference between the observations and the model equivalent state. Thus, the cost function Eq. (1) can be written in an incremental form as follows:

$$J = \frac{1}{2}(\delta c_0)^T \mathbf{B_c}^{-1}(\delta c_0) + \frac{1}{2}\sum_{i=0}^{n-1}(\delta e_i)^T \mathbf{B_{e_i}}^{-1}(\delta e_i) + \frac{1}{2}\sum_{i=0}^{n}(d_i - \boldsymbol{H_i}\delta c_i)^T \mathbf{R_i}^{-1}(d_i - \boldsymbol{H_i}\delta c_i) \tag{3}$$

Using a linearization approximation, Eq. (2) becomes

$$\delta c_i = L_{i,i-1}\delta c_{i-1} + L_{i,i-1}\Gamma_{i-1}\delta e_{i-1} \tag{4}$$

where $L_{i,i-1}$ and $\Gamma_{i-1}$ are Jacobians of $f_{i,i-1}$ with respect to $\delta c_{i-1}$ and $\delta e_{i-1}$, and $i = 1,2,\cdots n$. Thus, with a time integration, Eq. (4) can be presented as:

$$\delta c_i = L_{i,0}\delta c_0 + \sum_{l=0}^{i-1} L_{i,l}\Gamma_l \delta e_l \tag{5}$$

where $L_{i,0}$ denotes the tangent linear model operator of the CTM acting on $\delta c_0$, and the subscript is the time step from $i$ to the initial time. $L_{i,l}\Gamma_l$ $(l = 0,1,\cdots i-1)$ is the operator acting on $\delta e_l$, and $\Gamma_l$ is an operator that converts emissions to concentrations.

There are several numerical algorithms available to minimize the cost function in Eq. (3) (Courtier et al., 1994; Li and Navon, 2001). For the algorithms to minimize Eq. (3) with large dimensions, the gradient of the cost function is required. The gradient with respect to $\delta c_0$ and $\delta e_i$ $(i = 0,1,\cdots,n-1)$ can be written as:

$$\frac{\partial J}{\partial \delta c_0} = \mathbf{B_c}^{-1}(\delta c_0) + \sum_{l=0}^{n} L_{l,0}^T \boldsymbol{H_l}^T \mathbf{R_l}^{-1}(d_l - \boldsymbol{H_l}\delta c_l) \tag{6}$$

$$\frac{\partial J}{\partial \delta e_i} = \mathbf{B_{e_i}}^{-1}(\delta e_i) + \sum_{l=i+1}^{n} \Gamma_i^T L_{l,i}^T \boldsymbol{H_l}^T \mathbf{R_l}^{-1}(d_l - \boldsymbol{H_l}\delta c_l) \quad (i = 0,1,\cdots,n-1) \tag{7}$$

Here, $L_{0,0} = I$ for $i = 0$, where $I$ is an identity matrix. A time window of 6 h is typically used in operational synoptic-scale numerical weather predictions. Since a SO$_2$ lifetime in a model grid is usually less than 6 h (Fioletov et al., 2015), we still use a window of 6-h ($n = 6$) in the experiments presented in the following sections.

Eq. (6) and (7) include three types of adjoint operators, that is, $\Gamma^T, L^T$ and $\boldsymbol{H}^T$, which are derived from the tangent linear model operator $\Gamma, L$, and observation operator $\boldsymbol{H}$, respectively. The tangent linear operators $\Gamma$ and $L$ from WRF-Chem are very complex and computational demanding, we simplify the CTM to focus on SO$_2$.

WRF-Chem is an online coupled air quality model with sophisticated and comprehensive physical and chemical processes. Focusing on SO$_2$, the governing equation for the concentration can be written as:

$$\frac{\partial c}{\partial t} = -u\frac{\partial c}{\partial x} - v\frac{\partial c}{\partial y} - w\frac{\partial c}{\partial z} + \frac{\partial}{\partial x}\left(K_x \frac{\partial c}{\partial x}\right) + \frac{\partial}{\partial y}\left(K_y \frac{\partial c}{\partial y}\right) + \frac{\partial}{\partial z}\left(K_z \frac{\partial c}{\partial z}\right) - \mathbf{e}^{-\Lambda}\frac{\partial c}{\partial t} - r\frac{\partial c}{\partial t} + V_m \frac{\rho_{air}}{\rho}\frac{\Delta S}{dz}e \tag{8}$$

where $c$ is the gas/aerosol concentration, and $u, v$, and $w$ denote the wind in $x, y, and\ z$ directions, respectively. Thus, the $u\frac{\partial c}{dx} + v\frac{\partial c}{dy} + w\frac{\partial c}{dz}$ is a transport term. $K_x, K_y$, and $K_z$ are turbulent exchange coefficient in $x, y$, and $z$ directions, respectively, based on $K$ theory of turbulence, and $\frac{\partial}{\partial x}\left(K_x\frac{\partial c}{\partial x}\right) + \frac{\partial}{\partial y}\left(K_y\frac{\partial c}{\partial y}\right) + \frac{\partial}{\partial z}(K_z\frac{\partial c}{\partial z})$ is the turbulent term. In the study, the horizontal grid spacing is 27 km, thus the $\frac{\partial}{\partial x}(K_x\frac{\partial c}{\partial x}) + \frac{\partial}{\partial y}\left(K_y\frac{\partial c}{\partial y}\right)$ can be neglected. But the vertical turbulence term

$(\frac{\partial}{\partial z}(K_z\frac{\partial c}{\partial z}))$ should be retained since the vertical grid spacing is generally less 200 m in the lower and middle layers. $\mathbf{e}^{-\Lambda}\frac{\partial c}{\partial t}$ denotes the wet deposition term, where $\Lambda$ is the loss rate (Grell and Dévényi, 2002) and $\mathbf{e}$ is the base of natural logarithms (=0.272), $r\frac{\partial c}{\partial t}$ is the chemical term, where $r$ is the chemical reaction rate of the species, and $V_m\frac{\rho_{air}}{\rho}\frac{\Delta S}{dz}e$ is the emission term, where $e$ denotes the emission source of the species. $V_m = 22.4\times10^{-3}$ m$^3$ mol$^{-1}$ is the molar gas volume, $\rho$ is the air density of the actual atmosphere (kg m$^{-3}$), $\rho_{air}$ is the standard air density indicating the molar volume, and $\Delta S$ is the grid

area.

From the simplified Eq. (8), the model operators of $\Gamma\ and\ L$ can be written as:

$$L = -u\frac{\partial c}{\partial x} - v\frac{\partial c}{\partial y} - w\frac{\partial c}{\partial z} + \frac{\partial}{\partial z}\left(K_z\frac{\partial c}{\partial z}\right) - \mathbf{e}^{-\Lambda}\frac{\partial c}{\partial t} - r\frac{\partial c}{\partial t} \qquad (9)$$

$$\Gamma = V_m\frac{\rho_{air}}{\rho}\frac{\Delta S}{dz}e \qquad (10)$$

Using tangent linear coding techniques, we could derive the code for the discretized tangent linear operators $L$ and $\Gamma$

(Eq. (9-10)) from the source code built in WRF-Chem. Once the source code is available for the tangent linear operators, we use the adjoint coding technique to derive the adjoint operator. The adjoint coding technique are detailed in Hoffman et al. (1992).

**2.3 Observational and background error covariances**

$\mathbf{R}_i$ in Eq. (1) is the observational error covariance for a set of observations ($y_i$), where $\mathbf{B_c}$ and $\mathbf{B}_{e_i}$ are the BECs for

the concentrations and emissions, respectively. In a DA system, $\mathbf{R}_i$ and BEC play important roles in successful assimilation. The observational errors include the measurement error (observed value error) and representative error (error of observation operator $\mathbf{H}$). The observation error $\varepsilon_{SO_2}$ is defined as below:

$$\varepsilon_{SO_2} = \sqrt{\varepsilon_r{}^2 + \varepsilon_o{}^2} \qquad (11)$$

where $\varepsilon_o$ is the measurement error, and $\varepsilon_r$ is the representative error. The measurement error $\varepsilon_o$ is the systematic

error generated during monitoring by the instrument at each environmental monitoring station. Therefore, the measurement error $\varepsilon_o$ of SO$_2$ observation in this study is 1.0 µg m$^{-3}$, similar to that reported by Chen et al. (2019).

The representative error $\varepsilon_r$ is caused by converting the model variable to the observation variable (Schwart et al., 2012) and can be expressed as:

$$\varepsilon_r = \gamma\varepsilon_o\sqrt{\frac{dx}{L}} \qquad (12)$$

where $\gamma$ is an adjustable parameter scaling $\varepsilon_o$. $\gamma = 0.5$ was used in accordance with that used in Dai et al., (2021). Furthermore, $dx$ is the grid spacing (27 km in this study) and $L$ is the radius of influence of an observation, which was taken as 2 km according to that reported by Chen et al. (2019). Then, $\varepsilon_r = 1.8$ µg m$^{-3}$ calculated from Eq. (12).

     BECs ($\mathbf{B}_c$ and $\mathbf{B}_{e_i}$ in Eq. (1)) are the error covariance matrices of SO$_2$ concentrations and emissions. Practically, the BEC is overly large for handling numerically. Thus, we followed the method used by Li et al. (2013) and Zang et al. (2016)

to simplify $\mathbf{B}$:

$$\mathbf{B} = \mathbf{DCD}^{\mathrm{T}} \tag{13}$$

     where $\mathbf{D}$ is the RMSE matrix and $\mathbf{C}$ is the correlation matrix.

     $\mathbf{C}$ can be simplified by the Cholesky factorization and Kronecker product method (Li et al., 2013) as:

$$\mathbf{C}^{\frac{1}{2}} = \mathbf{C}_x^{\frac{1}{2}} \otimes \mathbf{C}_y^{\frac{1}{2}} \otimes \mathbf{C}_z^{\frac{1}{2}} \tag{14}$$

For $\mathbf{B}_{e_i}$, the standard deviation $\mathbf{D}_{e_i}$ is diagonal with a 200% error (Wang et al., 2012) and $\mathbf{C}_{e_i}$ is a block diagonal matrix, with the main diagonal blocks being the correlation matrices of SO$_2$ emission. The main diagonal blocks of $\mathbf{C}_{e_i}$ is 1.0 because the emission in each grid point is independent of that in other grids.

     The National Meteorological Center method (Parrish and Derber, 1992) was used to estimate the BEC of SO$_2$ concentrations. The differences between 48 h and 24 h forecasts were generated from 17 January 2020 to 18 February 2020.

The first initial chemical field at 0000 UTC on 17 January 2020 was obtained from a 10-d forecast in consideration of spin-up. The subsequent initial chemical fields were derived from the former forecast one day prior. The horizontal length scale was used to determine the magnitude of SO$_2$ variance in the horizontal direction. This scale can be estimated by the curve of the horizontal correlation with distances, and the horizontal correlation is approximately expressed by a Gaussian function

$\mathbf{e}^{\frac{(x1-x)^2}{2L_s^2}}$ ($\mathbf{e}$ is the base of natural logarithms equal to 0.272). Here, $x1$ and $x$ are two points, and $L_s$ is the horizontal length

scale. According to Zang et al. (2016), when the intersection of the decline curve reaches $\mathbf{e}^{1/2}$, the distance can be approximated as the horizontal length scale in Fig 2(a). The horizontal length scale was 81 km in this study. The vertical variance of SO$_2$ concentrations was considered by the vertical correlations in the BEC. A strong relationship was observed in the boundary layer (approximately below the 20th model layer) in the vertical direction (Fig. 2(b)). The standard deviation demonstrates the reliability of the forecasting model, and the standard deviation for the vertical distribution of SO$_2$

concentrations decreased with increasing height in the $\mathbf{B}_c$ (Fig. 2(c)).

        (a)                          (b)                         (c)

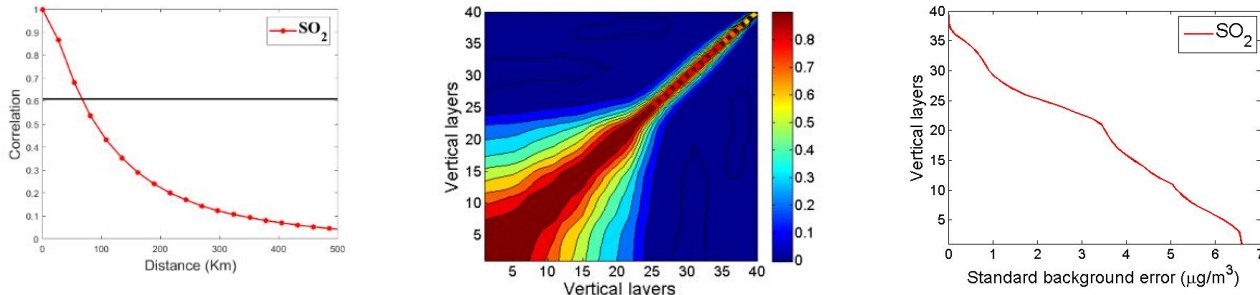

**Figure 2: Background error covariation of SO₂ concentrations. (a) Vertical distribution of the horizontal correlation; the horizontal thin black line is the reference line (e$^{1/2}$) used to determine the horizontal correlation scales. (b) Vertical correlations. (c) Vertical distribution of the standard deviation.**

### 2.4 Observation and emission data

Hourly SO₂ data obtained from the website of the China National Environmental Monitoring Center (http://www.cnemc.cn) were used for assimilation and evaluation. There were 1933 stations in China in January 2020. Most observational stations were located in central and eastern China, whereas the stations in the west were relatively sparse. The sites were gridded into the model grid ($27 \times 27$ km$^2$). If more than two stations were in the same grid, one station was randomly selected to verify the improvements relating to using optimized emissions, and the remaining sites were used for

assimilation. In this study, 508 sites were selected for verification, and the remaining 1425 stations were used for assimilation. A strict criterion was used to remove SO₂ observations with values exceeding 650 µg m$^{-3}$ to ensure data quality (Chen et al., 2019).

    The background anthropogenic emissions data were obtained from the MEIC (http://www.meicmodel.org/) developed by Tsinghua University, with a 0.25° × 0.25° resolution and 2016 as the base year. The MEIC is a "bottom-up" emission

inventory that covers 31 provinces on the Chinese mainland, and includes eight major chemical species (Zhang et al., 2009) and counts anthropogenic emissions from sources in five sectors (power, industry, residential, transportation, and agriculture). Details of the technology-based approach and source classifications has been reported by Zhang et al. (2009). The actual emission inventory (0.25 °× 0.25 °) was pre-processed to match the model grid spacing (27 km).

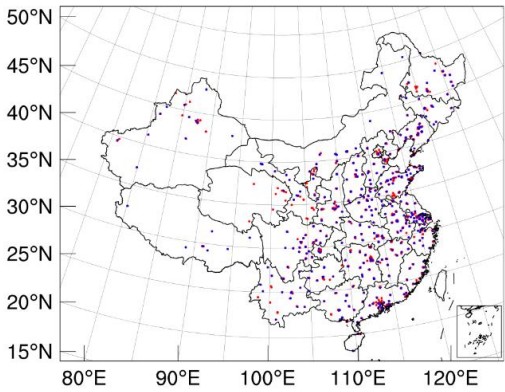

**Figure 3: Locations of the 1425 SO₂ assimilation observation stations (red) and 508 independent observation stations (blue). Inset: South China Sea.**

## 2.5 Experimental design

Figure 4 shows a flowchart of the procedure used to optimize SO₂ emissions in a single time step of $i$. First, a forecast was performed using the WRF-Chem model and background emissions to generate the meteorological and chemical fields, which were recorded every 10 min and then used in the 4DVAR system. Then, the 4DVAR system performed every 6 h to obtain SO₂ optimized emissions and initial concentrations by assimilating the hourly SO₂ observations. For example, the observations during 0000–0600 UTC were assimilated using Eq. (1). The assimilated SO₂ concentration initial field (0000 UTC) and the optimized SO₂ emissions during 0000–0500 UTC were obtained.

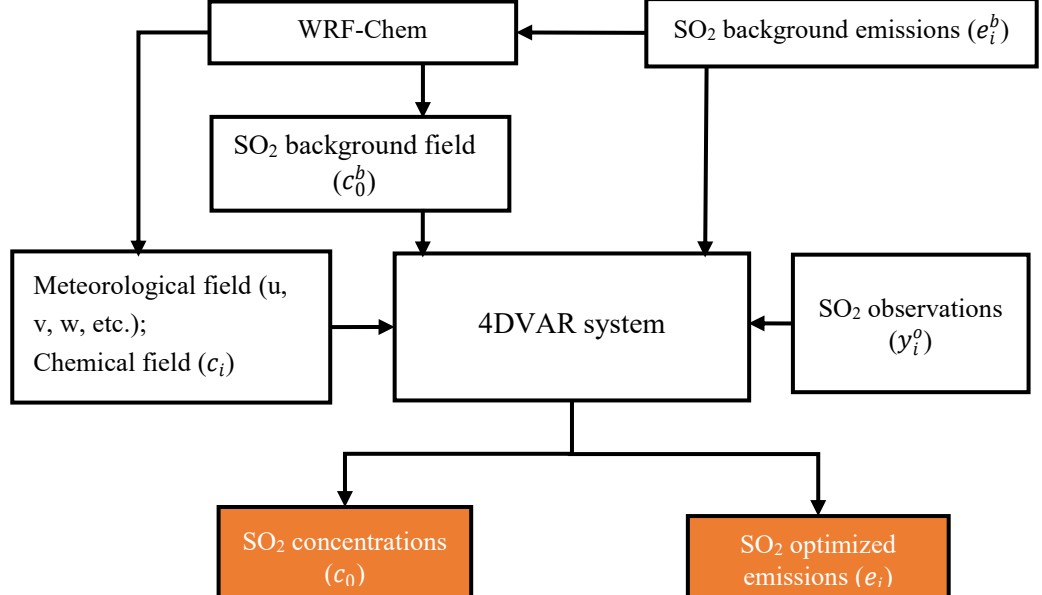

**Figure 4: Flow chart of the SO₂ emissions optimization procedure in a single time step of $i$. The orange boxes represent the SO₂ optimized emissions and SO₂ concentrations of output. The $c_0^b$, $c_0$, $e_i^b$ $e_i$ and $y_i^o$ are the mathematical symbols from Eq. (1).**

The SO₂ emissions during the COVID-19 lockdown over China were optimized to evaluate the performance of the 4DVAR system, and to analyse the reduction of SO₂ emissions related to the COVID-19 lockdown. The National lockdown was imposed in Wuhan and surrounding cities of Hubei provinces on 23 and 24 January 2020, respectively. Then, Chinese mainland implemented the national lockdown policies on 26 January 2020. We selected the study period from 17 January to 6 February 2020, which covered the time before and during lockdown. The latest available MEIC emission inventory is based on the statistics of 2016. However, the changes of emissions between 2020 and 2016 are both related to the emissions reduction policies and COVID-19 lockdown. The difference between 2019 and 2020 emissions during the same period reflected the influence of COVID-19 lockdown on SO₂ emissions. Thus, the SO₂ emissions during the study period in 2019 was also optimized.

Table 2 summarizes the details of DA emissions experiments. For the set of Emi_2019 experiments, the first DA process started on 17 January 2019, and the observations during 0000–0600 UTC of 17 January 2019 were assimilated by the 4DVAR system. Then, the optimized initial SO₂ concentration field (0000 UTC) and SO₂ emissions during 0000–0500 UTC were obtained. Before conducting Emi_2019 experiment, 24 h forecasts were performed by WRF-Chem with MEIC_2016 emissions every 0000 UTC from 17 January to 7 February, 2019 to provide the physical and chemical parameters. The daily chemical initial conditions were obtained from the 24 h forecast of the previous day. For the 24 h forecast, the meteorological initial and boundary conditions were provided by the 1° × 1° National Centers for Environmental Prediction (NCEP) Global Final Analysis data at a 6-hour frequency. The chemical boundary fields were not considered because the domain used in this study was wider than that in China. For the Emi_2019 experiment, the emissions of 2019 were optimized by the 4DVAR system every 6-h with the background emissions of MEIC_2016. The physical and chemical parameters used in this DA process were obtained from the WRF-Chem forecast. For the Emi_2020 experiment, the DA process settings were similar to those of the Emi_2019 experiment. The optimized emissions for 2020 were obtained with the emission 2019 as background emission.

**Table 2: Details of 4DVAR experiments to optimize emissions for 2019 and 2020**

| Name | Background emissions | Optimized emissions | Study period |
|------|----------------------|---------------------|--------------|
| Emi_2019 | MEIC_2016 | 2019 optimized emissions | Every 6 h from 17 January to 7 February, 2019 |
| Emi_2020 | 2019 optimized emissions | 2020 optimized emissions | Every 6 h from 17 January to 7 February, 2020 |

To estimate the improvement of SO₂ forecasts using optimized emissions, three sets of forecast experiments were performed using the MEIC_2016 emissions and the optimized emissions for 2019 and 2020, respectively, labeled Ctr_2016, DA_2019, and DA_2020, respectively (see Table 3). The three experiments were run daily with 24 h forecasts from 17 January to 7 February 2020 using the same WRF-Chem domain settings and physiochemical parameters. The SO₂ initial

condition (IC) at 0000 UTC on January 17 was based on the spin-up forecasts initialized at 0000 UTC on January 7, 2020 for all three forecast experiments. The SO$_2$ ICs were later obtained from the 24-h forecast of the previous day for the three

experiments, respectively. For example, the SO$_2$ IC of the experiment beginning at 0000 UTC on 18 January was taken from the 24-h forecast result of the experiment beginning at 0000 UTC of 17 January, and so on. Meteorological initial and boundary conditions were provided by the 1° × 1° NCEP Global Final Analysis data at a 6-h frequency. The chemical boundary fields were not considered.

**Table 3: Details of the forecast experiments using emissions from 2016, 2019 and 2020.**

| Name | Emission | Forecast duration | Study period |
|---|---|---|---|
| Ctrl_2016 | MEIC_2016 | 24 h | Daily from 17 January to 7 February, 2020 |
| DA_2019 | The 2019 optimized emissions | 24 h | Daily from 17 January to 7 February, 2020 |
| DA_2020 | The 2020 optimized emissions | 24 h | Daily from 17 January to 7 February, 2020 |

**3 Results**

**3.1 Results of 4DVAR emission experiments**

3.1.1 4DVAR test case

The first day (17 January 2019) was used as a test case to determine the effectiveness of using 4DVAR. The experiment employed MEIC_2016 as the background emissions and assimilated the hourly surface SO$_2$ observations during 0000–0600

UTC of 17 January 2019. The observed SO$_2$ concentrations in Fig. 5a indicated the heavy polluted areas with SO$_2$ concentrations exceeding 80 μg m$^{-3}$ were mostly located in the North China Plain and Northeast China. The areas lightly polluted with SO$_2$ concentrations below 40 μg m$^{-3}$ were mostly located in Southern China. Compared with the observed concentrations, the background concentrations (Fig. 5b) were underestimated in North China Plain and Northeast China but overestimated in Central China and the Sichuan Basin. Figure 5c shows the increment field of SO$_2$ concentrations (analyzed

field minus background field). Positive values in most of Northern China and negative in Central China and the Sichuan Basin were observed, suggesting that the optimized IC is more consistent with the observed SO$_2$ concentrations than the background concentrations. The evaluations of the optimized IC and background concentrations are shown in Fig. 5d. Compared with the background field, the mean bias in analysis field improved from –2.76 to 1.79 μg m$^{-3}$and RMSE decreased from 23.12 to 11.81 μg m$^{-3}$ and the correlation coefficient (CORR) of analysis field increased from 0.19 to 0.84.

The result indicates that the accuracy of the ICs of SO$_2$ concentrations were improved after using the 4DVAR method. The forecast accuracy can be improved using optimized ICs (Peng et al., 2017, 2018), but the emission is the most important factor influencing the forecast accuracy. The emissions and IC concentrations were simultaneously optimized in the EMI_2019 experiment using our 4DVAR system.

Figure 5e presents the background emission of MEIC 2016 at 0000 UTC. According to Fig. 5a and 5b, MEIC_2016

emissions underestimated in most of Northern China and overestimated in Central China and Sichuan Basin. Fig. 5f shows

the increment of SO₂ emissions at 0000 UTC 17 January 2019 by using the 4DVAR system. There were positive increment in North China Plain and Northeast China, and negative increment in Central China and Sichuan Basin. The distribution of the incremental SO₂ emissions was consistent with that of the incremental SO₂ concentration (Fig. 5c). There is a reasonable relationship between the two increments since the underestimated/overestimated emission may result in underestimated/overestimated simulation of concentration.

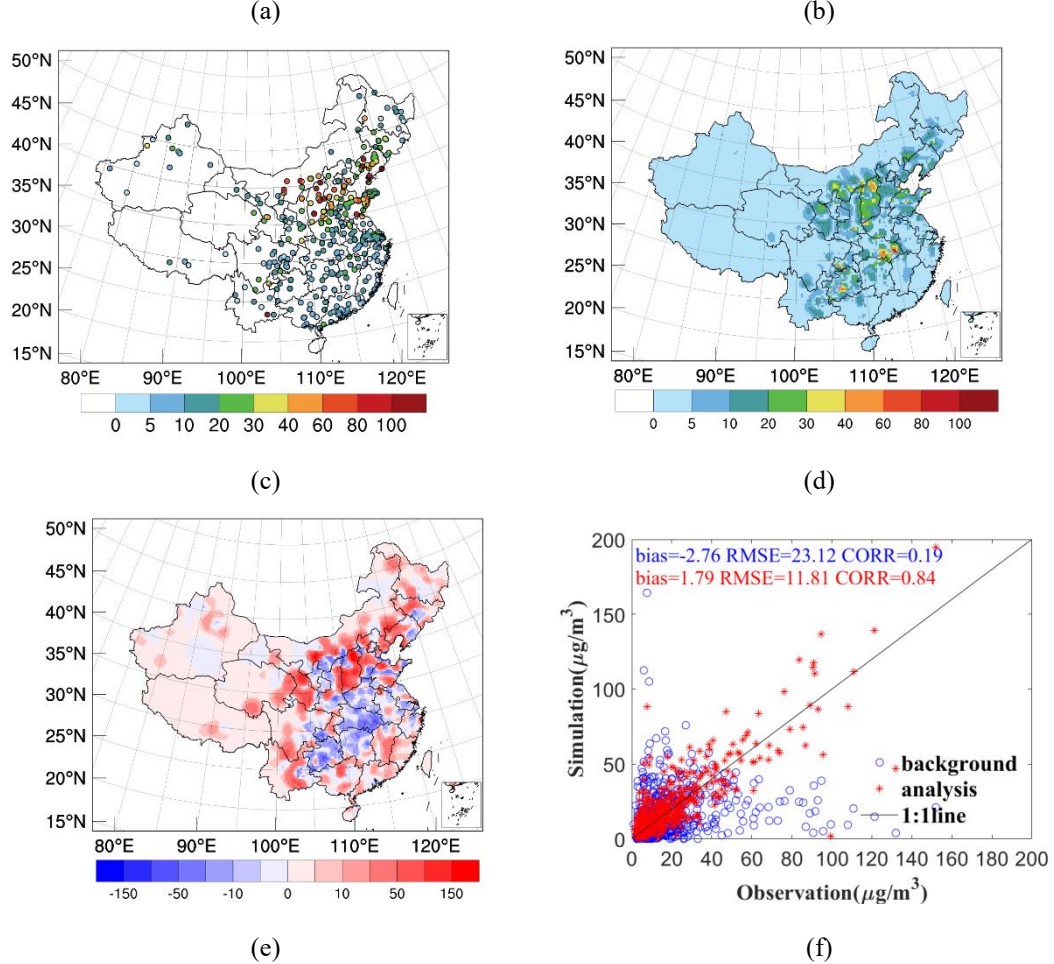

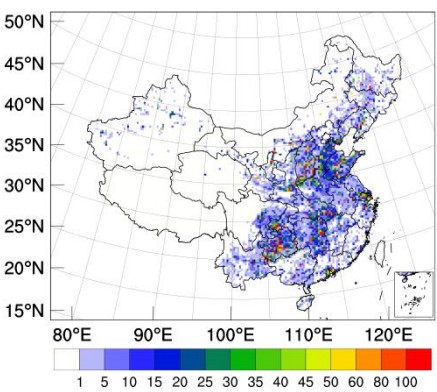
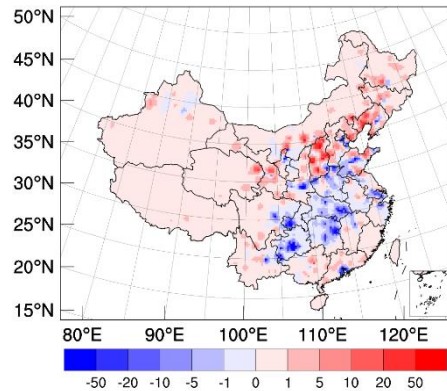

**Figure 5: Simulated and observed SO₂ concentrations at 0000 UTC 17 January 2019. (a) Observations, (b) background concentrations, (c) SO₂ concentrations increment, (d) scatter plots, (e) background emissions, and (f) SO₂ emissions increment. Units: μg m⁻³ for (a), (b), (c), and (d), and mol km⁻² h⁻¹ for (e) and (f). Insets: South China Sea.**

**3.1.2 Spatial distribution of emission**

Compared with MEIC_2016 (Figure 6a), the emissions for 2019 and 2020 (Fig. 6b and 6c) from the 4DVAR experiments of Emi_2019 and Emi_2020 decreased in the area with heavy emissions, particularly in North China Plain and Central China. The reduction in emissions between 2019 and MEIC_2016 may primarily result from the national pollution control policy. However, the reduction of emissions between 2020 and 2019 may primarily result from the COVID_2019

lockdown, including school and workplace closures, event and public gathering cancellation, and restrictions on public transport.

Figure 7a shows the difference in emissions between 2020 and 2019. The negative values were seen in most of the areas with strict lockdown, such as North China Plain, Central China, Yangtze River Delta, and Pearl River Delta. It indicates that the 2020 emission substantially decrease, compared with the 2019 emission due to the COVID-19 lockdown. The reducing

ratio of emission was averaged in China as 9.2% (Fig. 7b), but over 40.0% in most areas of North China and Central China. Zheng et al. (2020) have found that SO₂ emissions in China decreased by 12.0% in January and February 2020 compared to values in 2019. Fan et al. (2020) have also reported the SO₂ concentration decreased by 20.0–50.0% over China during the COVID-19 lockdown period in the spring of 2020 based on TROPOMI satellite data. Our results are similar to those of previous studies. In addition, SO₂ emissions increased in some areas of Northeast China, Tibetan Plateau, Yunnan Province,

and the southeast coastal areas, where the epidemic was weaker than that in other areas (Kraemer Moritz et al., 2020; Tian et al., 2020). Most of the increase in SO₂ was <10 mol km⁻² h⁻¹, but the positive ratios were >100.0%, suggesting that new emission sources were generated. It is suggested that these newly generated emissions were probably due to relocating power plants and factories from cities to the surrounding villages (Chen et al., 2019).

(a)                                          (b)                                          (c)

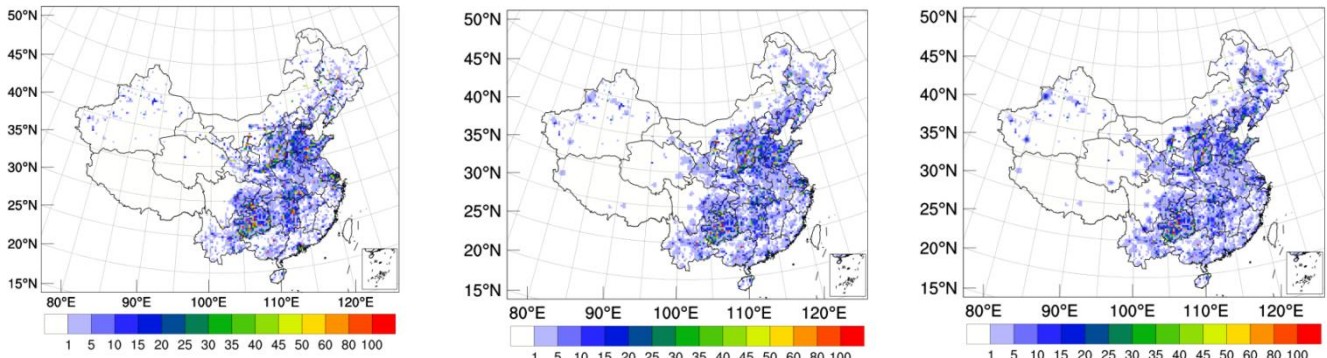

**Figure 6: Emissions in China for (a) MEIC_2016, (b) 2019 and (c) 2020. Units: mol km⁻² h⁻¹. Insets: South China Sea.**

(a)              (b)

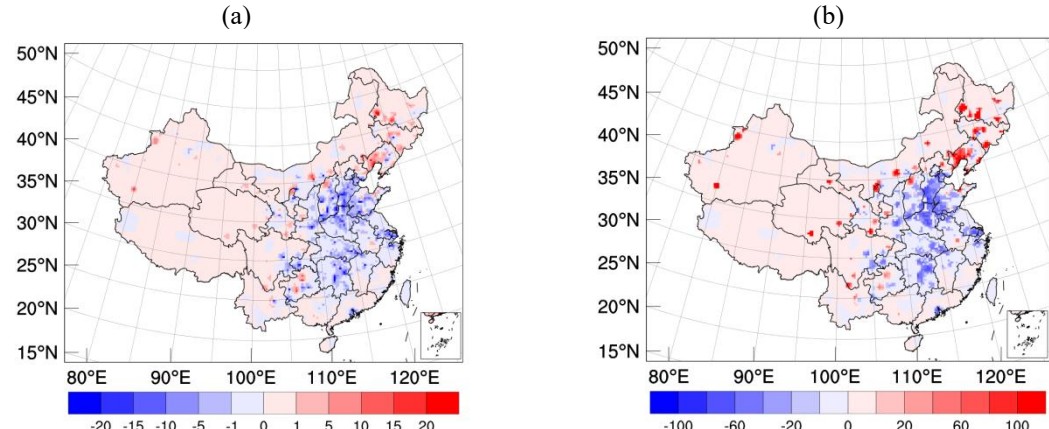

**Figure 7: (a) Difference between 2020 and 2019 emissions and (b) ratios of (2020–2019)/2019 emissions in China. Units are mol km⁻² h⁻¹ for (a) and percent (%) for (b). Insets: South China Sea.**

Figure 8 shows the similar analyses as in Fig. 6, but for Central China. Wuhan first implemented the first-level response to the COVID-19 with strict lockdown policies on 23 January 2020, and the entire Hubei province implemented lockdown on 24 January 2020. The heavy emissions exceeding 20.0 mol km⁻² h⁻¹ were most located around large cities from the emissions of 2019 and 2020 (Fig. 8b and 8c). Figure 9a shows the difference between 2020 and 2019 emissions in Central China. The average emission value in Wuhan was 43.0 mol km⁻² h⁻¹ in 2019 and 34.0 mol km⁻² h⁻¹ in 2020, showing a reduction of 21.0% compared with the emissions for 2019. Al-qaness et al. (2021) have also found approximately 15% decrease in $SO_2$ concentrations with 15% around Wuhan. Furthermore, almost all emissions around the large cities decreased by 5–10 mol km⁻² h⁻¹ (Fig. 9a), and the negative ratios were >20.0% (Fig. 9b). The large reduction in $SO_2$ emissions were related to the decrease in industrial and domestic coal combustion and power plants during the COVID-19 lockdown (Zheng et al., 2018, 2020; Bian et al., 2019; van der A et al., 2017).

(a)              (b)              (c)

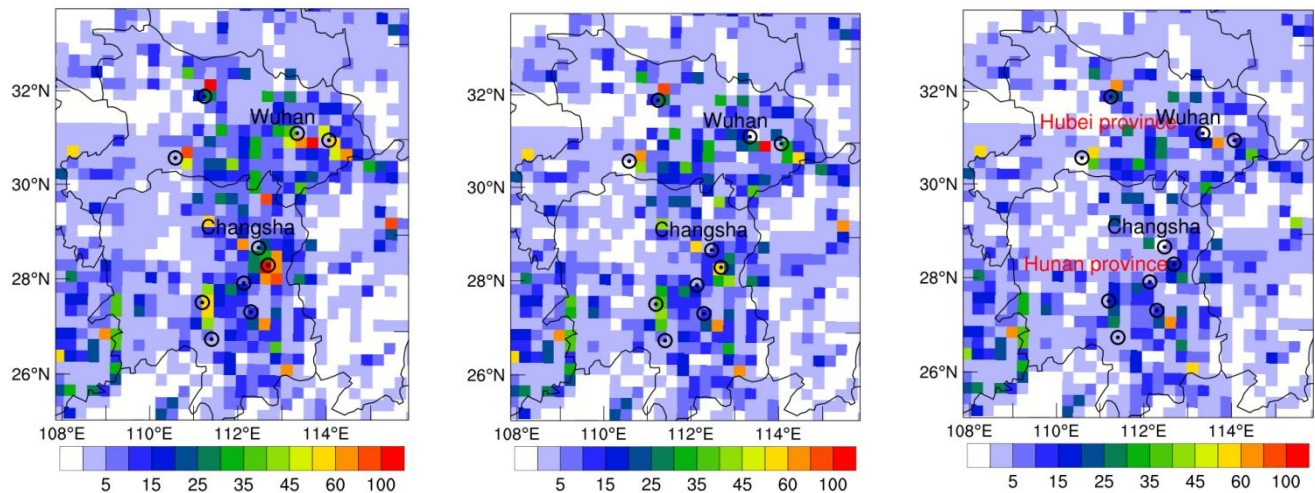

**Figure 8:Emissions in Central China for (a) MEIC_2016, (b) 2019, and (c) 2020. Black circles with dots are the locations of large cities. The red characters in (c) are the name of provinces, and the black characters are the name of cities. Wuhan and Changsha are the capitals of Hubei and Hunan provinces, respectively. Unit: mol km$^{-2}$ h$^{-1}$.**

(a)                                                                          (b)

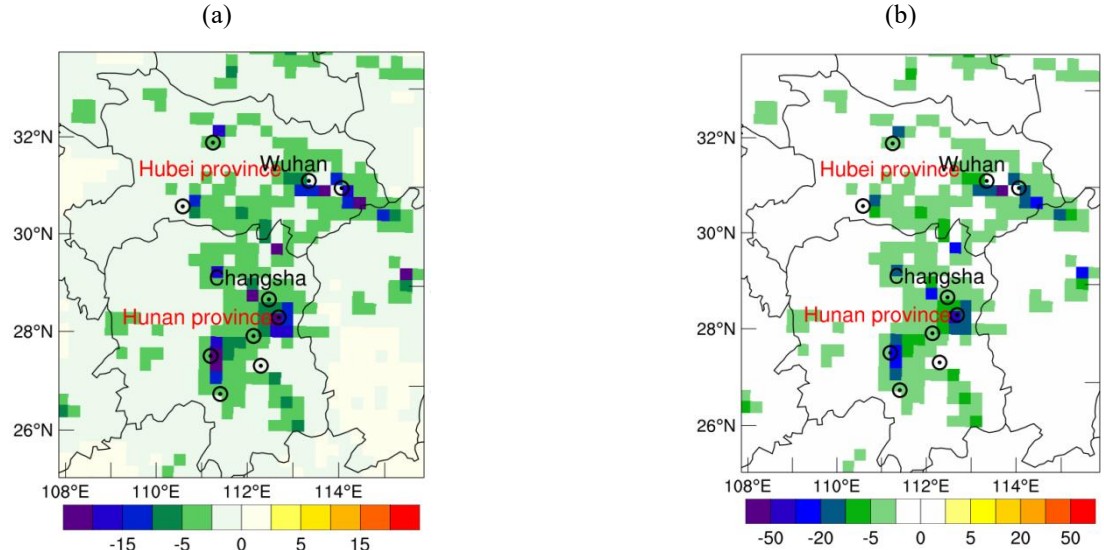

**Figure 9: (a) Difference between 2020 and 2019 emissions and (b) ratios of (2020–2019)/2019 emissions in Central China. Black circles with dots are the location of large cities. The red characters are the name of provinces, and the black characters are the name of cities. Wuhan and Changsha are the capitals of Hubei and Hunan provinces, respectively. Units: mol km$^{-2}$ h$^{-1}$ for (a) and percent (%) for (b).**

### 3.1.3 Temporal evolution of emissions

Figure 10 shows the daily $SO_2$ emissions for MEIC_2016, Emi_2019, and Emi_2020 over all grid points. The average emissions in Chinese mainland (Fig. 10a) from MEIC_2016, Emi_2019, and Emi_2020 were $42.2\times10^6$, $40.1\times10^6$, and $36.4\times10^6$ kg d$^{-1}$ during the same period from 17 January to 7 February. The emissions for 2020 decreased by 9.2% compared with those for 2019, indicating a decrease between 2020 and 2019 due to the COVID-19 related lockdown. In Emi_2019 emissions, the lowest emissions occurred on 1 February 2019, but increased during 4–6 February, 2019 mainly attributed to the traditional firework displays during Spring Festival (Wang et al., 2007; Zhang et al., 2020; Huang et al., 2021a). Complex changes in $SO_2$ emission trends were observed in 2020 in relation to reduced human activity. For example, a peak of $40.1 \times 10^6$ kg d$^{-1}$ occurred on 24 January, 2020, in relation to firework displays (Fig. 10a), after which the $SO_2$ emissions decreased because of the COVID-19 lockdown. For Central China, the average $SO_2$ emissions were $5.7\times10^6$, $4.2\times10^6$ and $3.1\times10^6$ kg d$^{-1}$ during the same period from 17 January to 7 February (Fig. 10b). The $SO_2$ emissions peaked at $3.5\times10^6$ kg d$^{-1}$ on 24 January, 2020 due to firework displays, and a reduction began from 26 January, 2020 because of the national lockdown.

(a)                                                            (b)

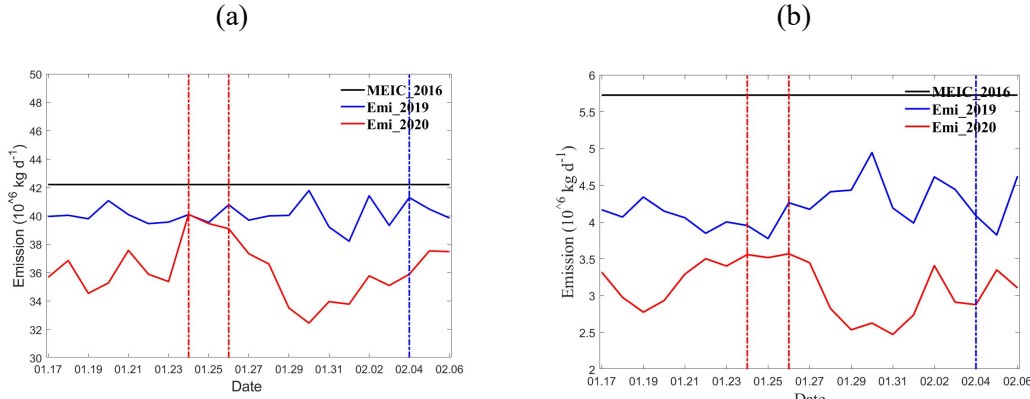

**Figure 10: Time series of daily $SO_2$ emissions in (a) China and (b) Central China. The red dotted lines represent the dates of the start of national lockdown and the Chinese Spring Festival in 2020. The blue dotted line represents the Chinese Spring Festival in 2019. Units: $10^6$ kg d$^{-1}$.**

Figure 11 shows the average hourly emissions for MEIC_2016, Emi_2019, and Emi_2020 emissions from 17 January to 7 February over all grid points. The hourly factors for MEIC_2016 were obtained from power plant report, with two peaks during the day at 0100 UTC (0900 BJ time – Beijing time) and 0900 UTC (1700 BJ time) to reflect the emissions at rush hours (Chen et al., 2019b; Hu et al., 2022). In the Chinese mainland (Fig. 11a), the emissions pf 2019 and 2020 were lower than those of MEIC_2016 during 0600–1200 UTC. This is primarily due to the recent implementation of China's emission reduction policies and the COVID-19 lockdown. Previous studies have shown that the second peak (0900 UTC) of $SO_2$ emissions had weakened (Chen et al., 2019), which was also reflected in our hourly emission analysis. The emissions for Emi_2019 and Emi_2020 were higher than those of MEIC_2016 during 1600–2000 UTC, but remain almost unchanged

between Emi_2019 and Emi_2020 emissions. During this time period, most factories were closed and human activities were reduced. The SO₂ emissions are primarily emitted from power plants, and the changes in emissions are small between different years (Zheng et al., 2018, 2021; Hu et al., 2021). Thus, the increase in Emi_2019 and Emi_2020 emissions during 1600–2000 UTC are mainly due to the uncertainties of MEIC_2016 (Chen et al., 2019). Compared with the average emissions for Emi_2019, those for Emi_2020 emissions decreased by 18.0%, reflecting the reduction due to the COVID-19

lockdown. The emissions in 2019 and 2020 in Central China were lower than those in MEIC_2016 for 24 h period, with a maximum reduction at 0900 UTC (Fig. 9b). Compared with the emissions in 2019, the emissions in 2020 appreciably decreased by 22.3–42.1%. The first peak of the emissions in 2020 was delayed and occurred at 0200 UTC because of the national lockdown policies. The most substantial reduction between 2019 and 2020 emissions was $-120.4 \times 10^3$ kg h$^{-1}$ at 01 UTC, reflecting the change in human activities at the first peak. Additionally, although there was only a moderate decrease

in SO₂ emissions ($-72.3 \times 10^3$ kg h$^{-1}$) at 1300 UTC, the reduction ratio ($-54.5\%$) was the largest during 24 h.

(a) (b)

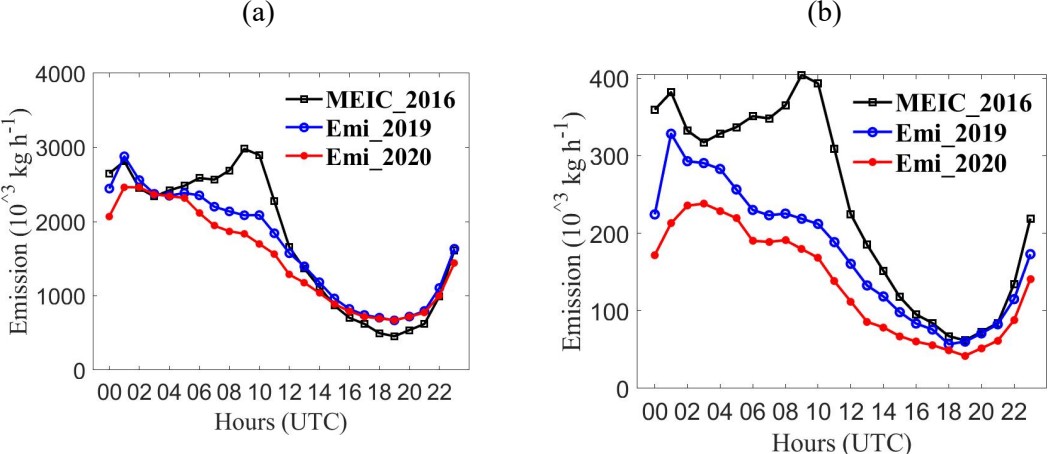

Figure 11: Hourly emissions for (a) China and (b) Central China (Unit: $10^3$ kg h$^{-1}$).

**3.2 Results of forecast experiments**

Using the emissions of MEIC_2016, Emi_2019 and Emi_2020, three forecast experiments (Ctrl_2016, DA_2019 and DA_2020 in Table 3) were implemented to further demonstrate the effect of optimized emissions. Figure 12 shows the

average 24-h forecast of SO₂ concentrations of the three forecast experiments over all stations in China during the study period from 17 January to 7 February 2020. The DA_2020 experiment with the 2020 emissions performed much better than the Ctrl_2016 and DA_2019 experiments, indicating that the emission is one of the most important factors for 24 h forecasts. The SO₂ concentrations in Ctrl_2016 and DA_2019 were overestimated, particularly during 0800–1800 UTC (Fig. 12a), while the SO₂ concentrations in DA_2020 are similar to the observed concentrations. The result showed the 4DVAR system

effectively optimize emissions and improve the accuracy of forecast. The average RMSEs of the three experiments were 21.7, 15.6, and 10.7 µg m$^{-3}$, respectively. Compared to the average RMSE of Ctrl_2016 experiment, the RMSEs of the

DA_2019 and DA_2020 decreased by 28.1% and 50.7%. The average CORRs for the Ctrl_2016, DA_2019, and DA_2020 experiments were 0.20, 0.38, and 0.61, respectively. Thus, the average CORRs for DA_2019 and DA_2020 experiments increased by 89.5% and 205.9% from the CORR for Ctrl_2016 experiment. The average bias of Ctrl_2016, DA_2019, and DA_2020 experiments were 5.9, 4.9, and –0.1 µg m$^{-3}$, respectively. It is suggested that the optimized emissions could substantially improve forecast accuracy, and the 4DVAR approach is effective to optimize daily and hourly emission during an accidental special event.

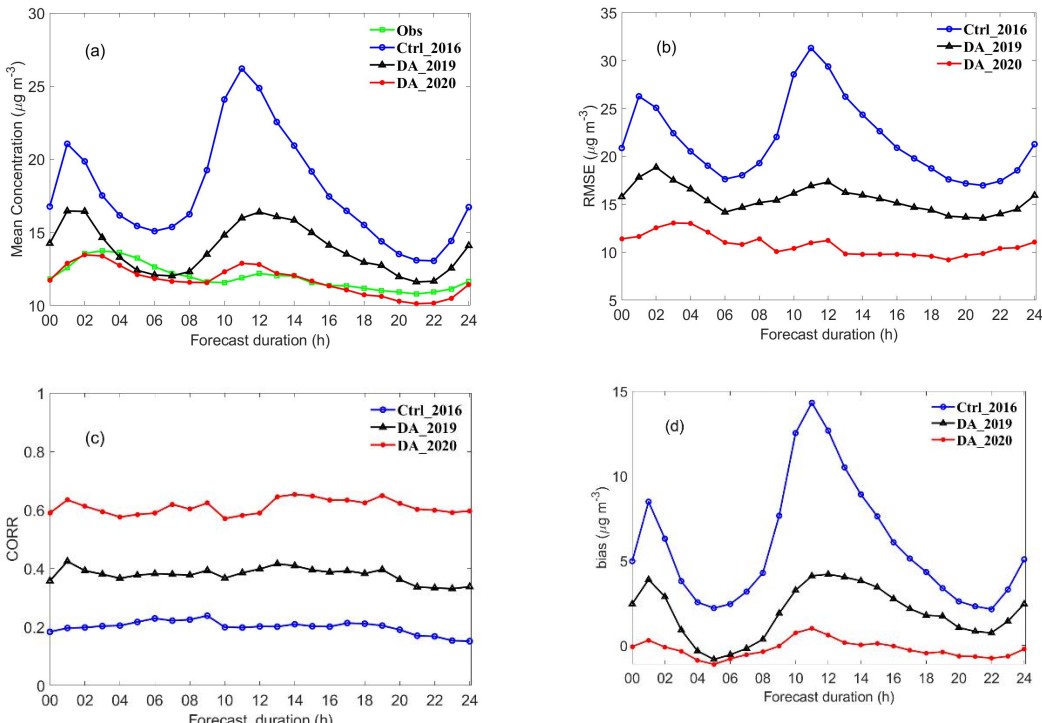

**Figure 12: Forecast accuracy of SO$_2$ concentrations in China for the Ctrl_2016, DA_2019, and DA_2020 experiments during the study period in 2020: (a) mean concentration, (b) RMSE, (c) CORR, and (d) bias. Unit: µg m$^{-3}$ for (a), (b), and (d). Obs: Observation.**

Figure 13 shows the same analyses as those presented in Fig. 12, but for Central China. It also showed that the forecast accuracies of the DA_2019 and DA_2020 experiments were higher than those of Ctrl_2016. The average observation concentration was <10 µg m$^{-3}$, which is substantially lower than that for Chinese mainland (Fig. 12a). The mean concentration of DA_2020 was close to the observed concentration in Central China. The above results suggest that although the 2020 optimized emissions were generally consistent with the real emissions, they were slightly higher than the real emissions. In the 4DVAR optimization process, each grid will be influenced by surrounding grids because of the advection and vertical mixing. The theory of 4DVAR method is to take a balance between the observations and background field and to obtain the optimized field. Therefore, when the observations are lower and the background field are higher, the value of the optimized field will be higher than the observation. Compared to that of Ctrl_2016, the average bias of the DA_2019 and

DA_2020 experiments decreased from 20.1 to 12.6 and 3.5 μg m$^{-3}$. The average RMSE decreased by 48.8% and 77.0%, and the average CORR increased by 44.3% and 238.7%. This indicates that the forecast accuracy substantially improved after using optimized emissions.

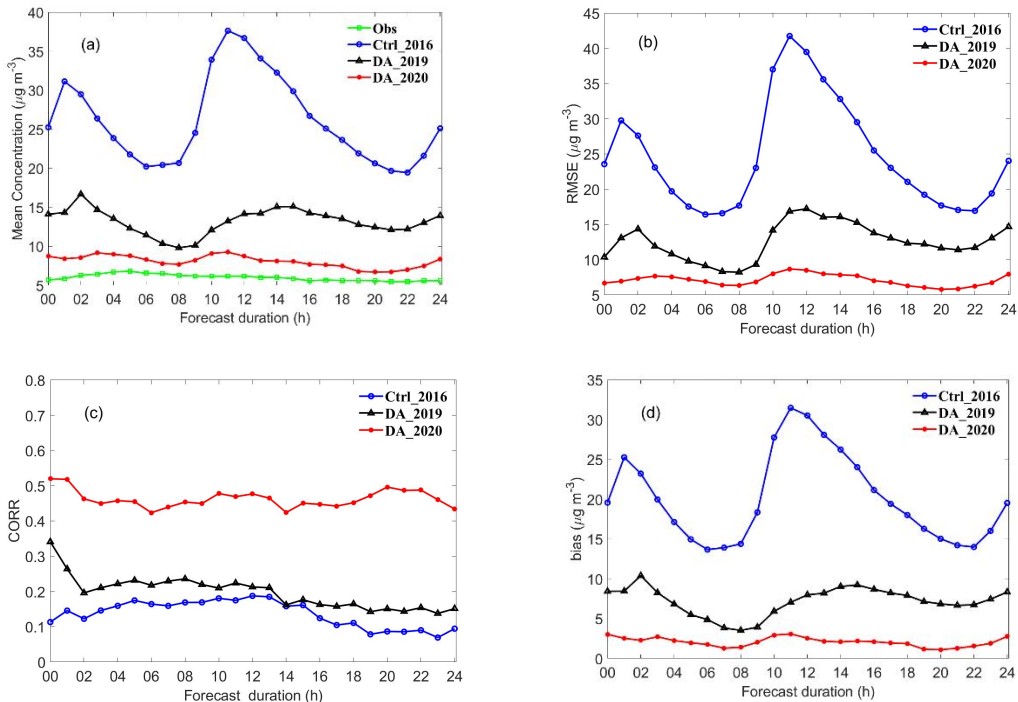

**Figure 13: Forecast accuracy of SO$_2$ concentrations in Central China using the Ctrl_2016, DA_2019, and DA_2020 experiments during the study period in 2020. (a) mean concentration, (b) root-mean-square error (RMSE), (c) correlation coefficient (CORR),**
**and (d) bias. Unit: μg m$^{-3}$ for (a), (b), and (d). Obs: Observation.**

## 4 Conclusions

  In this study, we developed a 4DVAR system based on the WRF-Chem model to estimate SO$_2$ emissions, where the initial SO$_2$ concentration and emissions were set as the state variables to estimate SO$_2$ emissions. An adjoint operator was derived from the WRF-Chem model, focusing on the processes of transport, dry/wet deposition, vertical turbulence, and SO$_2$
chemical reactions. Hourly SO$_2$ concentration observations were assimilated to optimize SO$_2$ emissions, which were used to improve the SO$_2$ forecasting accuracy.

  The 4DVAR system was applied to investigate SO$_2$ emission changes during the COVID-19 lockdown in China, particularly focusing on Central China. The MEIC_2016 emissions were set as the background values. The average emissions of MEIC_2016, 2019, and 2020 were 42.2×10$^6$, 40.1×10$^6$, and 36.4×10$^6$ kg d$^{-1}$, namely 2020 emissions decreased
by 9.2% compared with those in 2019, indicating a substantial decrease between 2019 and 2020 due to the COVID-19

related lockdown. The average 2020 emissions in Central China dropped by 21.0% compared to the 2019 emissions, owing to the strict lockdown policy during COVID-19. The largest decrease in emissions occurred in Wuhan (decline of 57.0%), which COVID-19 had heavily affected by this time. Hourly average emissions were analyzed to estimate the changes between 2019 and 2020. Compared with 2019 emissions, the average 2020 emissions decreased by 18.0%, reflecting
lockdown-associated reduction in $SO_2$ emissions. The 2020 emissions in Central China decreased by 22.3–42.1% compared with the 2019 emissions.

Three sets of forecast experiments for 2020, using MEIC_2016, Emi_2019, and Emi_2020 emissions, were conducted to illustrate the effects of the optimized emissions. The experiment with MEIC_2016 emissions overestimated the $SO_2$ concentration forecast, whereas the experiment with 2019 optimized emissions decreased the concentrations but still
overestimated the values. The forecast accuracy of the experiment with the 2020 emissions was the closest to the observation. The RMSE of the experiments with the emissions in 2019 and 2020 decreased from 21.7 to 15.6, and 10.7 μg m$^{-3}$, respectively, and the correlation coefficient increased from 0.20 to 0.38 and 0.61, respectively, compared with those of the experiment with MEIC_2016 emissions. For Central China, the average RMSE and correlation coefficient of the experiment with MEIC_2016 were 24.6 μg m$^{-3}$ and 0.1. Compared with the average RMSE of the experiment with MEIC_2016, those of
the experiments with 2019 and 2020 emissions decreased by 48.8% and 77.0%, and the average correlation coefficient increased by 44.3% and 238.7%.

Though our 4DVAR system could effectively optimize real time emission as a "top-down" approach, some limitations still remain. Only hourly surface $SO_2$ observations were used to constrain the emission sources. The spatial distribution of surface observation sites was uneven with fewer sites in the northwest and southwest regions, resulting in limited
adjustments to emission sources in these regions. In future, satellite data will be used to adjust the emission source to address the lack of surface observation data. Furthermore, the simultaneous optimization of $SO_2$ concentrations and emissions will be implemented in a 4DVAR system, and multi-source observation data will be used to improve its performance.

**Author contributions:** Zengliang Zang designed the overall research; Yiwen Hu performed experiments; Yanfei Liang, Wei You and Xiaobin Pan contributed to the development of the DA system; Zengliang Zang and Xiaoyan Ma provided funds; Yiwen Hu, Zengliang Zang, and Xiaoyan Ma. wrote the paper, with contributions from all co-authors; Xiaoyan Ma, Zengliang Zang and Zhijin Li developed the mathematical formulation and reviewed the paper. All authors have read and agreed to the published version of the manuscript.

**Funding:** This research was funded by the National Natural Science Foundation of China (Grant Nos. 41975167, 42061134009, 41775123,
41975002, and 42061134009). This research was supported by the National Key Scientific and Technological Infrastructure project "Earth System Science Numerical Simulator Facility" (EarthLab).

**Data availability:** The data and data analysis method are available upon request.

**Competing interests:** The authors declare that they have no conflict of interest.

**Acknowledgments:** NCEP FNL reanalysis data were downloaded from https://rda.ucar.edu/datasets/ds083.2/, last access: 13 April 2022.
The MEIC 2016 emission sources were developed by Tsinghua University (http://meicmodel.org/?lang=en, last access: 13 April 2022). The hourly $SO_2$ observations were downloaded from the CNEMC website (http://www.cnemc.cn, last access: 13 April 2022).

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
