# Peer review of "Four-dimensional Variational Assimilation for SO2 Emission and its Application around the COVID-19 lockdown in the spring 2020 over China"

_Atmospheric Chemistry and Physics, 2022_

## Referee Comment (RC2)

**Review of**

**Four-dimensional Variational Assimilation for SO2 Emission and its Application around the COVID-19 lockdown in the spring 2020 over China**

by

**Yiwen Hu et al.**

**Overall comments**

This manuscript describes a four-dimensional variational (4DVAR) data assimilation (DA) system designed to update SO2 emissions by assimilating surface SO2 concentrations. The system was applied to study emissions reductions over China during the beginning of the COVID-19 pandemic. Additionally, the authors showed how obtaining SO2 emissions from 4DVAR can potentially lead to better subsequent SO2 concentration forecasts relative to using static SO2 emissions.

I think the topics in this manuscript are interesting and important, and I have few concerns with the overall objectives of this work. However, I think there are shortcomings regarding descriptions of the DA system and the experiments, most of which are related to lack of clarity. In addition, I don't think the OSSE part of the manuscript is necessary, and I instead suggest replacing this material with other metrics demonstrating that the 4DVAR DA system works as expected. Finally, the writing is mostly understandable, but there are many grammatical errors that should be fixed prior to publication in ACP.

**Major comments**

- 1. I think more descriptions about the mechanisms through which your 4DVAR DA system updates SO2 emissions are needed. It isn't entirely obvious how this adjustment is achieved using 4DVAR. Did you somehow model cross-covariances between SO2 concentrations and emissions? Or is the tangent-linear/adjoint model effectively altering the emissions? I was left unsure of exactly how you updated SO2 emissions, and I think this key point of your method needs to be explicitly described. It might also be worth contrasting your method with how an EnKF works; in the EnKF it would be straightforward to update SO2 emissions, as SO2 emissions could simply be added to the state vector and be naturally updated by ensemble-based covariances.
- 2. I believe that when harvesting synthetic observations from "truth" in an OSSE it is customary to add noise drawn from an "observation error" distribution to the synthetic observations. However, your methods don't mention doing this, which makes me wonder if the OSSE was properly performed. By not adding noise to harvested observations, you will likely get analysis fits much closer to "truth" than if you add noise to the synthetic observations you eventually assimilate.

Regardless, I don't feel the material about the OSSE adds much to the manuscript, and it is little more than a "sanity check" that I didn't find very convincing. I think a better

demonstration of the 4DVAR system's efficacy would be to show plots of: 1) The cost function reduction from a 4DVAR analysis that assimilated real observations; 2) Analysis increments directly showing how SO2 emissions were modified by assimilating SO2 observations; and 3) "observation minus background" and "observation minus analysis" statistics. I think presenting these types of plots would increase confidence that your 4DVAR DA system works as expected and can provide stronger evidence of system robustness than the OSSE.

Additionally, I'm somewhat concerned with Fig. 5b, which shows that the cost function increased between the 8th and 9th iterations. Variational cost functions must monotonically decrease with each inner-loop iteration, so there is a chance that something went wrong. Please look into this or offer an explanation.

3. I think section 2.4 needs to be more specific about what you actually did, rather than making fairly general statements. What specific observation error did you use? What were your values of  $\varepsilon_r$  and  $\varepsilon_o$  and how did you arrive at these values?

Similarly, in lines 201-204, please state how you produced these 48- and 24-h forecasts. What model configuration did you use? You should also cite the "NMC method" (Parrish and Derber 1992) for this approach. Moreover, in line 202, should "state variables" be "background errors"? How did you model the correlations in *C*, especially for the emissions? Overall, please be more specific about your background and observation error covariance construction.

- 4. Several aspects of the experimental design were not initially clear to me and caused confusion. Although some elements became clearer with time, I think descriptions of the experiments should be clarified:
  - a. From Table 3, it appears that you performed DA experiments for ~3 weeks. What was the cycling period of your experiments (i.e., how often did you produce new 4DVAR analyses)? Was it 6 hours? Did you continuously cycle both the chemistry and meteorology, or did you periodically update meteorology from an external source, like the GFS model? What did you do for chemical boundary conditions (my apologies if I missed it)? Furthermore, Table 3 states 24-h forecasts were produced, but how often did you initialize these 24-h forecasts? Overall, the temporal aspect of the experiments should be clarified.
  - b. Fig. 3 didn't seem clear to me. What specific field(s) are being updated? Just SO2 emissions? Or both SO2 emissions and concentrations? Additionally, this figure might be clearer if you annotated the mathematical symbols from Eq. (1) or (3) on it so readers can link this figure to the equations. It might also be nice if you added another panel to the figure showing the temporal progression of the DA system (per above comment). Finally, in the top-left box, there's a typo (it should be "field").

c. It appears that you ran the 4DVAR DA system over two separate periods to estimate SO2 emissions: 1) 17 Jan – 7 Feb 2019; and 2) 17 Jan – 7 Feb 2020. However, you never explicitly stated this! Thus, there are really two parts to this work. The first is estimating emissions in 2019 and 2020 from the 4DVAR DA system. The second is using different emissions estimates to drive various sets of forecasts over a common period in 2020. This distinction was not always clear, which caused me confusion.

Please explicitly state these various experiments and their purposes. It is important that you do so because of places like line 328, where you simply stated the *years* of the emissions, and not the names in Table 3 (which contain years); more distinction needs to be made for data *for a given year* versus various experiments *in 2020* that used emissions generated from various years.

- d. For the DA\_2019 experiment, you effectively seemed to use a "pre-processing" step, where you ran the 4DVAR DA system over 2019 and then used those emissions when simulating a period in 2020. It might be worth noting that in contrast, the DA\_2020 experiment did everything all at once without the need for a "pre-processing" step.
- 5. Section 3.2: Please clarify that the data (i.e., emissions) discussed in this subsection were obtained from the 4DVAR analyses (I think), and not some other source. In general, please be more precise about from where the data on each figure come from.
- 6. The writing is understandable but there are many grammatical errors that I found distracting. Please carefully proofread the manuscript.

**Minor comments**

- 1. Line 48: "most explored algorithms" for what? Please be specific.
- 2. Lines 50-51: The second instance of "to estimate" should be removed, and perhaps "total regional and global emissions" should be moved to after the first instance of "to estimate".
- 3. Lines 60-63: This statement is too broad. There are techniques to handle this problem, like inflation, that are well established at least for meteorological EnKF DA. Please refine this statement.
- 4. Line 89: Please add a reference for WRF-Chem.
- 5. Line 105: Suggest "covering all of China" instead of "covering the entire country".
- 6. Lines 105, 158, 159: "resolution" should be "grid spacing", as the two are not the same.
- 7. Lines 111, Table 1: Please add a reference for the Grell-3D scheme.
- 8. Fig. 1 caption: Please state that this figure also shows the WRF modeling domain!
- 9. Lines 122-125: Somewhere in here, please specifically define *n*.
- 10. Somewhere in section 2.2, please be more precise about which "control variables" are included in *c*0. Is it just SO2 concentration?
- 11. Line 131: Here, is *H* nonlinear? (probably it is). Please state.
- 12. Lines 131-132: Please state that *R* is the observation error covariance matrix.

- 13. Line 133: Please define *f*.
- 14. Eq. (3): Some more explanation is needed about how you go from Eq. (1) to Eq. (3) for the observation term. Specifically, please note the linearization about the background.
- 15. Lines 150-152: How does this relate to the cycling period of the DA system? Does this mean you produced new analyses every 6?
- 16. Line 161: "large horizontal resolution study" is unclear. Are you referring to your specific study or something else? Please clarify.
- 17. Eq. (10):  $L_{turb}$  doesn't appear in the list of quantities in line 169, and  $L_{dry}$ , which does appear in line 169, doesn't appear in the list of equations. Please clarify. Also please double check Eq. (14).
- 18. Lines 170-185: Should  $L_{trub}$  be  $L_{turb}$ ?
- 19. Line 180: Should it be Eqs. 9–13 instead of Eqs. 9–12?
- 20. Lines 168-185: I'm not an expert about adjoint modeling, but I had the feeling that these lines aren't precise enough about the adjoint model formulation. Shouldn't there be more derivatives in there?
- 21. Line 192: Please omit "the assimilation variable, which is the"...it's confusing, because that phrase is somewhat referring to a state/control variable, even though you're really talking about observation errors.
- 22. In lines 220-228, please be very precise about "emission" vs. "concentration" in your descriptions.
- 23. Line 243: Please change February 6 to February 7 for consistency with Table 3.
- 24. Line 246: Typo: it should be "physiochemical".
- 25. Line 249: Suggest "...based on the spin-up forecasts initialized at 0000 UTC...".
- 26. Line 250: Please be more precise about "the previous day" (this comment relates to earlier comments about the cycling period).
- 27. Line 256: There seem to be more than 13 x 9 points in Fig. 4, so I was confused about this statement concerning "arrays and columns". Please clarify.
- 28. Fig. 5: The legend in the left panel is covering data and should be moved, and the y-axis in the right panel should probably be "J" not "Jb".
- 29. Line 282: I'm not sure I agree with this statement, especially in (a) and (b); the 2019 SO2 concentration decreases with time but the SO2 emissions seem steady. Please revise.
- 30. Line 283: To my eyes, it looks like the lowest emissions were on 1 February, not 3 February (per Fig. 6b).
- 31. Lines 289-292: Please point to Figs. 6c,d here.
- 32. Fig. 6 caption: Please state the meanings of the vertical lines.
- 33. Line 311: I believes "rates" should be "ratios".
- 34. Fig. 7: What are the insets in the lower right corner of each panel? Additionally, please be more precise about the subtraction convention. Above (c) and (d), it says "2020 2019" but the caption says, "differences between 2019 and 2020", which implies "2019 2020". It might be clearest to just write out "2020 minus 2019". Finally, please state in the caption whether these statistics are averaged over the entire period. Similar comments also apply to Fig. 8.
- 35. Line 315: I don't think "observations" is the correct word. Is "analyses" more accurate? Please also see line 363.
- 36. Line 320: Please remove "slightly"; it's too subjective.

- 37. Line 321: Please remove "Remarkably", which is also subjective, and furthermore, the differences don't seem "remarkable".
- 38. Line 333: Please remove "slightly". Also, it seems that this behavior was only evident in Fig. 9a, so please clarify the region you are discussing.
- 39. Fig. 9 caption: Are these statistics averaged/aggregated over the entire period and over all sites or grid points? Please clarify.
- 40. Lines 352-355: I found this chunk troublesome. The explanation you offered didn't make sense to me, and I'm not sure all your statements are accurate. Please clarify or omit.
- 41. Throughout, including figure captions: "Skill" should be "accuracy". Skill is "accuracy relative to a baseline", and all of the metrics you are showing are measures of accuracy, not skill. I believe every instance of "skill" needs to be changed to "accuracy".
- 42. Fig. 10 caption: Please clarify whether these statistics are averaged/aggregated over the entire time period and all sites. Same comment for Fig. 11.
- 43. Lines 369-370: Please omit "compared with the Ctrl 2016 experiment".
- 44. Lines 373-374: Please clarify what you mean by the "background field". Do you mean the field at the very start of the period (0000 UTC 17 January 2020)?
- 45. Line 390: Can you point to a figure for this key result about the decrease of optimized emissions? Also, did you ever state these values in the results section (sorry if I missed it)?
- 46. Figs. 1, 4, 5, 8, 9: Please add annotations (e.g., "a", "b") to all these figures.

---

## Author Comment (AC1)

**Response to the Comments of Referees**

**Manuscript ID: acp-2022-301**

**Title:** Four-dimensional Variational Assimilation for SO2 Emission and its Application around the COVID-19 lockdown in the spring 2020 over China

**Author:** Yiwen Hu, Zengliang Zang\*, Xiaoyan Ma\*, Yi Li, Yanfei Liang, Wei You, Xiaobin Pan, Zhijin Li

We thank the reviewers and editors for providing helpful comments to improve the manuscript. We have revised the manuscript according to the comments and suggestions of the referees.

The referee's comments are reproduced (black) along with our replies (blue). All

the authors have read the revised manuscript and agreed with submission in its revised form.

**< Anonymous Referee #1 >**

**Comment:** This manuscript describes the development and application of a 4DVAR system to optimize SO2 emissions in China. An OSSE test shows improved consistency with the true emissions after optimizing emissions using this system. The framework has also been applied to estimate SO2 emissions during the COVID19 shutdown and shows a reduction of 18% compared to 2019. The topic fits the readership of ACP. I recommend publication after addressing the following comments:

**Response:** We thank the referee for the positive comments on our manuscript. The manuscript has been carefully revised according to the referee's comments and suggestions.

Comment 1: L18, please specify the studied region in the abstract.

**Response 1: Corrected.**

**Comment 2:** L64 - 77, I would expect literature reviews on the application of 4D-Var to SO2 emission estimates in this paragraph. There are several of such studies. How are these 4D-Var estimates compared with previous EnKF SO2 estimates and your results?

**Response 2:** Thanks for your suggestion. For the 4DVAR method, Qu et al. (2019) estimated SO2 emissions based on the GEOS-Chem model and its adjoint model by assimilating OMI observations and found the SO2 emissions decreased by 48% over China from 2008 to 2016. For the EnKF method, many studies estimated SO2 emissions by assimilating surface and satellite observational data in recent years, such as Dai et al (2021), Chen et al. (2019), Koukouli et al. (2018) and so on. Dai et al. (2021) developed a four-dimensional local ensemble transform Kalman filter and showed that the SO2 emissions over China in November 2016 decreased by 49.4% in comparison to the 2010 background emission due to national pollution control policies (Zheng et al., 2018).

Above literature review has been added in the introduction and discussion of revised manuscript.

**Comment 3:** Eq (1), it was not clear to me whether H is an operator or a matrix from my first glance. I suggest using a different font for H.

**Response 3:** H is an observation operator that computes the observation estimates from the state variables and is also a vector. Additionally, the H in our 4DVAR system is Linear. Thus we use an italic bold font H to represent the operator in the revise manuscript.

**Comment 4:** Eq (3), the use of H delta(c) here implies that the operator is linear, but I doubt that for SO2. Could you discuss the impact of this assumption on the results?

**Response 4:** In this study, H plays a role of interpolation from the model grid to the observed value and is linear. And the tangent linear operators  $\Gamma$ , L(in Eq. (6) and (7)) were applied to calculate  $c_i$ (or  $\delta c_i$ ) in this study.  $\Gamma$ , and L are derived from WRF-Chem are very complex and computational demanding, thus, we simplify the CTM to focus on SO2.

Focusing on SO2, the governing equation for the concentration of species in WRF-Chem can be written as:

$$\frac{\partial c}{\partial t} = -u\frac{\partial c}{\partial x} - v\frac{\partial c}{\partial y} - w\frac{\partial c}{\partial z} + \frac{\partial}{\partial x}\left(K_x\frac{\partial c}{\partial x}\right) + \frac{\partial}{\partial y}\left(K_y\frac{\partial c}{\partial y}\right) + \frac{\partial}{\partial z}\left(K_z\frac{\partial c}{\partial z}\right) - \mathbf{e}^{-\Lambda}\frac{\partial c}{\partial t} - r\frac{\partial c}{\partial t} + V_m\frac{\rho_{air}\Delta S}{\rho}\frac{\Delta S}{dz}e$$
(1)

where *c* is the gas/aerosol concentration,  $\partial x$  and  $\partial y$  are the horizontal resolutions of the model,  $\partial z$  is the vertical resolution, and *u*, *v*, and *w* denote the wind in *x*, *y*, and *z* directions, respectively.  $K_x$ ,  $K_y$  and  $K_z$  are turbulent exchange coefficient in *x*, *y*, and *z* directions based on K theory of turbulence.  $\Lambda$  is the loss rate. **e** is the base of natural logarithms equal to 0.272. *r* is the chemical reaction rate of the species. *e* denotes the emission source of the species.  $V_m = 22.4 \times 10^{-3} \text{ m}^3 \text{ mol}^{-1}$  is the molar volume of the gas,  $\rho$  is the air density of the actual atmosphere (kg m-3),  $\rho_{air}$  is the standard air density, and  $\Delta S$  is the grid area.

According to Eq. (1), the change in concentrations is approximately linear because the concentrations only relate to the physicochemical parameters, such as  $u, v, w, K_z, \Lambda$ ,  $r, \rho$ , and e.

We used the values  $(u, v, w, K_z, \Lambda, r, \rho, \text{ and } e)$  within an integration step (10mins) to represent the mean of these variables in the 4DVAR system. This process would lead some errors due to the linear operators. But even we used a shorter integration step of 2mins, the result is close to that of the integration step of 10mins (Fig. R1). The average difference in concentrations between the two experiments was 0.3 in the grid of i = 94, j = 152 during 1 hour (Fig. R1a), which was also 1% of the total average concentrations. The mean difference in concentrations over China was 0.1 (Fig.R1b). Thus, it is concluded that the error from the linearization is very small and negligible.

Figure R1: The forecast concentrations in the forward process by using different values in: (a) the grid of i = 94, j = 152 and (b) China.

**Comment 5:** L239, it is not clear to me what is the objective of these experiments just based on what is described here. Please clarify.

**Response 5:** Thank you for your suggestion. The goal of this study is to estimate the influence of COVID-19 lockdown on  $SO_2$  emissions. But, the newest emission inventory of MEIC is for 2016 by statistics. It is considerable inaccurate due to the emissions reduction policies. The difference between 2019 and 2020 emissions during the same period reflected the influence of COVID-19 lockdown on  $SO_2$  emissions. Thus, the  $SO_2$  emissions during the study period in 2019 was also optimized.

Table R1 shows the details of DA emissions experiments. For the set of experiments of Emi 2019, the first DA process started on 17 January 2019, and the observations during 0000-0600 UTC of 17 January 2019 were assimilated by the 4DVAR system. Then, the optimized SO2 concentration initial field (0000 UTC) and the optimized SO2 emission during 0000-0500 UTC were obtained. Before conducting Emi 2019 experiment, 24 h forecasts were performed by WRF-Chem with MEIC 2016 emissions every 0000 UTC from 17 January to 7 February, 2019 to provide physical and chemical parameter. The chemical ICs of each day were obtained from the 24 h forecasting of the previous day. For the 24 h forecast, the meteorological initial and boundary conditions were provided by the  $1^{\circ} \times 1^{\circ}$  National Centers for Environmental Prediction (NCEP) Global Final Analysis data at a 6-hour frequency. The chemical boundary fields were not considered because the domain used in this study was wider than China. For the experiment of Emi 2019, the emission of 2019 were optimized by 4DVAR system every 6 hours with the background emissions of MEIC 2016. The physical and chemical parameter used in this DA process were obtained by the WRF-Chem forecast. For the experiment of Emi 2020, the DA process settings are similar with the Emi 2019 experiment. The optimized emissions of 2020 is obtained with the emission 2019 as background emission.

Table R1: Details of 4DVAR experiments to optimize the emission for 2019 and 2020.

| Name     | Background emissions        | Optimized emissions         | Study period                                        |
|----------|-----------------------------|-----------------------------|-----------------------------------------------------|
| Emi_2019 | MEIC_2016                   | 2019 optimized
emissions | Every 6 hours from17 January to 7
February, 2019 |
| Emi_2020 | 2019 optimized
emissions | 2020 optimized
emissions | Every 6 hours from17 January to 7
February, 2020 |

On the base of three emission inventories of 2016, 2019 and 2020, three sets of forecast experiments were performed on the emissions during COVID-19 from 17 January to7 February 2020 (Table R2) to estimate the improvement of SO2 forecasts using optimized emissions. Three experiments were run daily with 24 h forecasts from 17 January to 7 February 2020, and all experiments used the same WRF-Chem domain settings and physiochemical parameters. The MEIC 2016 emissions were used in the Ctrl 2016 experiment. For the DA 2019 experiment, the 2019 optimized emissions were used to simulate SO2 concentrations during the study period. For the DA 2020 experiment, the 2020 optimized emissions were applied. The  $SO_2$  initial condition at 0000 UTC on January 17 was based on the spin-up forecasts initialized at 0000 UTC on January 7, 2020 for all three forecast experiments. The SO2 ICs were later obtained from the 24h forecasting of the previous day for the three experiments, respectively. For example, the SO2 IC of the experiment beginning at 0000 UTC on 18 January was from the 24h forecast result of the experiment beginning at 0000 UTC on 17 January, and so on. Meteorological initial and boundary conditions were provided by the  $1^{\circ} \times 1^{\circ}$  NCEP Global Final Analysis data at a 6-hour frequency. The chemical boundary fields were not considered.

| Name      | Emission                     | Forecast duration | Study period                                  |
|-----------|------------------------------|-------------------|-----------------------------------------------|
| Ctrl_2016 | MEIC_2016                    | 24 h              | Every day from 17 January to 7 February, 2020 |
| DA_2019   | The 2019 optimized emissions | 24 h              | Every day from 17 January to 7 February, 2020 |
| DA_2020   | The 2020 optimized emissions | 24 h              | Every day from 17 January to 7 February, 2020 |

Table 3: Details of the forecast experiments with emissions of 2016, 2019 and 2020.

**The description of experiments has been revised in the revised manuscript.**

**Comment 6:** Fig 7 & 8, how are these emission changes compared with previous studies?

**Response 6**: Figure 7 in the original manuscript shows the spatial differences and variations in emission ratios between 2019 and 2020. Compared with the 2019 optimized emissions, the 2020 optimized emissions decreased over most areas of the country due to the lockdown. The averaged reducing ratios of emission were 9.2% over China. Especially, the reducing ratios were more than 40.0% in most areas of North China and Central China. Zheng et al. (2020) showed that SO2 emissions in China decreased by 12.0% in January and February 2020 compared to that in 2019. Fan et al. (2020) also found the SO2 concentration decreased by 20.0–50.0% around COVID-19 lockdown in the spring 2020 over China based on TROPOMI satellite data. Our results were similar with the previous studies.

Figure 8 shows the same analysis as Fig. 7, but for central China. The averaged emission value in Wuhan was 43.0 mol km-2 h-1 in 2019 and 34.0 mol km-2 h-1 in 2020, showing a reduction of 21.0%. Almost all emissions in the Hubei Province decreased by 5–10 mol km-2 h-1. Al-qaness et al. (2021) also found a decrease of SO2 concentrations with 15% around Wuhan. In addition, the heavy emissions with the value exceeding 20.0 mol km-2 h-1 were most located around large cities. These heavy emissions decreased more than –5 mol km-2 h-1, and the negative ratios were >20.0%. The large reduction of SO2 emissions were coming from industrial and domestic coal combustion and power plants decreased during the COVID-19 lockdown (Zheng et al., 2018, 2020; Bian et al., 2019; van der A et al., 2017).

These sentences has been added in the revised manuscript.

**Comment 7:** Fig 11, the observation is significantly smaller than the simulations, even after DA. Could you address this a bit more and discuss the implications of this? How is this compared to other studies?

**Response 7:** Thank you for your suggestion. Figure 11a in the original manuscript shows the observation is smaller than the simulations in Central China. Note that the

mean concentration from the simulation of DA\_2020 is close to the concentration of observation, suggesting that the 2020 optimized emissions were generally consistent with the real emissions in 2020. In 4DVAR optimization process, each grid will be influenced by surrounding grids because of the advection and vertical mixing. The theory of 4DVAR method is to take a balance between the observation and background field and obtain the optimized field. Therefore, when the observation values are small and those of the background field are large, the value of the optimized field will be larger than the observation.

The discussion had been added in the revised manuscript.

**Reference**

Al-qaness, M. A. A., Fan, H., Ewees, A. A., Yousri, D., and Abd Elaziz, M.: Improved ANFIS model for forecasting Wuhan City Air Quality and analysis COVID-19 lockdown impacts on air quality, Environmental Research, 194, 110607, https://doi.org/10.1016/j.envres.2020.110607, 2021.

Chen, D., Liu, Z., Ban, J., and Chen, M.: The 2015 and 2016 wintertime air pollution in China: SO2 emission changes derived from a WRF-Chem/EnKF coupled data assimilation system, Atmos. Chem. Phys., 19, 8619-8650, 10.5194/acp-19-8619-2019, 2019.

Dai, T., Cheng, Y., Goto, D., Li, Y., Tang, X., Shi, G., and Nakajima, T.: Revealing the sulfur dioxide emission reductions in China by assimilating surface observations in WRF-Chem, Atmos. Chem. Phys., 21, 4357-4379, 10.5194/acp-21-4357-2021, 2021.

Qu, Z., Henze, D. K., Theys, N., Wang, J., and Wang, W.: Hybrid Mass Balance/4D-Var Joint Inversion of NOx and SO2 Emissions in East Asia, Journal of Geophysical Research: Atmospheres, 124, 8203-8224, https://doi.org/10.1029/2018JD030240, 2019.

Zheng, B., Tong, D., Li, M., Liu, F., Hong, C., Geng, G., Li, H., Li, X., Peng, L., Qi, J., Yan, L., Zhang, Y., Zhao, H., Zheng, Y., He, K., and Zhang, Q.: Trends in China's anthropogenic emissions since 2010 as the consequence of clean air actions, Atmospheric Chemistry and Physics, 18, 14095-14111, 10.5194/acp-18-14095-2018, 2018.

Zheng, B., Zhang, Q., Geng, G.-n., Chen, C., Shi, Q., Cui, M., Lei, Y., and He, K. J. E. S. S. D.: Changes in China's anthropogenic emissions and air quality during the COVID-19 pandemic in 2020, 2021.

---

## Author Comment (AC2)

**Response to the Comments of Referees**

**Manuscript ID:** acp-2022-301

**Title:** Four-dimensional Variational Assimilation for $SO_2$ Emission and its Application around the COVID-19 lockdown in the spring 2020 over China

**Author:** Yiwen Hu, Zengliang Zang[*], Xiaoyan Ma[*], Yi Li, Yanfei Liang, Wei You, Xiaobin Pan, Zhijin Li

We thank the reviewers and editors for providing insightful and constructive comments, which help improve the presentation of the manuscript. We have revised the manuscript according to the comments and suggestions of the referees.

The referee's comments are reproduced (black) along with our replies (blue). All the authors have read the revised manuscript and agreed with submission in its revised form.

**< Anonymous Referee #2 >**

**Overall comments**

This manuscript describes a four-dimensional variational (4DVAR) data assimilation (DA) system designed to update $SO_2$ emissions by assimilating surface $SO_2$ concentrations. The system was applied to study emissions reductions over China during the beginning of the COVID-19 pandemic. Additionally, the authors showed how obtaining $SO_2$ emissions from 4DVAR can potentially lead to better subsequent $SO_2$ concentration forecasts relative to using static $SO_2$ emissions.

**Response:** We thank the referee for the positive assessment of our manuscript. The manuscript has been carefully revised according to the referee's comments and suggestions.

I think the topics in this manuscript are interesting and important, and I have few concerns with the overall objectives of this work. However, I think there are shortcomings regarding descriptions of the DA system and the experiments, most of

which are related to lack of clarity. In addition, I don't think the OSSE part of the manuscript is necessary, and I instead suggest replacing this material with other metrics demonstrating that the 4DVAR DA system works as expected. Finally, the writing is mostly understandable, but there are many grammatical errors that should be fixed prior to publication in ACP.

**Response:** Thank you for your comments and suggestions. Following your advice, we have made systematic revisions on description of the DA algorithm, and the OSSE part has been removed. The 4DVAR system's efficacy due to the assimilation of real observations has been discussed to show the effectiveness of the DA system in emission optimization.

We have carefully corrected grammatical errors in the revised manuscript.

Below are our point-to-point response to your comments in detail.

**Major comments**

**Commemt1:** I think more descriptions about the mechanisms through which your 4DVAR DA system updates $SO_2$ emissions are needed. It isn't entirely obvious how this adjustment is achieved using 4DVAR. Did you somehow model cross-covariances between $SO_2$ concentrations and emissions? Or is the tangent-linear/adjoint model effectively altering the emissions? I was left unsure of exactly how you updated $SO_2$ emissions, and I think this key point of your method needs to be explicitly described. It might also be worth contrasting your method with how an EnKF works; in the EnKF it would be straightforward to update $SO_2$ emissions, as $SO_2$ emissions could simply be added to the state vector and be naturally updated by ensemble-based covariances.

**Response 1:** Thanks for your questions and suggestions. The cross-covariances is used for the EnKF method to update $SO_2$ emissions by assimilating $SO_2$ concentrations. But for the 4DVAR method, it is not necessary to use the cross-covariances between $SO_2$ concentrations and emissions. In the 4DVAR system, the $SO_2$ emission is the state vector and can be directly updated by assimilating $SO_2$ concentration

observations.

In our 4DVAR system, the cost function is as follows:

$$J = \frac{1}{2}\left(c_0 - c_0^b\right)^T \mathbf{B}_c^{-1}\left(c_0 - c_0^b\right) + \frac{1}{2}\sum_{i=0}^{n-1}\left(e_i - e_i^b\right)^T \mathbf{B}_{e_i}^{-1}\left(e_i - e_i^b\right) + \frac{1}{2}\sum_{i=0}^{n}\left(y_i^o - \boldsymbol{H_i}c_i\right)^T \mathbf{R}_i^{-1}\left(y_i^o - \boldsymbol{H_i}c_i\right)$$

(1)

where $c_0$ is the state variable that denotes the initial concentration vector, and $e_i$ is also the state variable that denotes the emission at each time step. The $c_0$ and the $e_i$ are assumed independent. Subscripts of the variables represent the time step. The $c_0$ at the initial time and the $e_i$ at the time steps from $0$ to $n-1$ are directly updated, when the optimal $J$ is obtained.

The SO$_2$ concentrations and the SO$_2$ emissions are connected by the differential equation of Eq. (4) in manuscript. The $c_i$ depend on the emission $e_{i-1}$. When we minimize the cost function of Eq. (1), the SO$_2$ emission is used in the calculation of the observation term of $\frac{1}{2}\sum_{i=0}^{n}(y_i^o - H_i c_i)^T R_i^{-1}(y_i^o - H_i c_i)$. Thus, the SO$_2$ emission can be updated by the observation of SO$_2$ concentration. Although both EnKF and 4DVAR methods are based on a model to show the relationship between concentration and emission, different ways are used in two methods. For the EnKF method, the cross-covariances between concentrations and emissions are generated by an ensemble of model outputs. But the 4DVAR method depends on the dynamical equations of model to establish the relationship between concentration and emission.

This statement has been added in the revised manuscript.

**Commemt2a:** I believe that when harvesting synthetic observations from "truth" in an OSSE it is customary to add noise drawn from an "observation error" distribution to the synthetic observations. However, your methods don't mention doing this, which makes me wonder if the OSSE was properly performed. By not adding noise to harvested observations, you will likely get analysis fits much closer to "truth" than if you add noise to the synthetic observations you eventually assimilate.

**Response 2a:** In the OSSE, the "real" emissions (EM_real) included 273 (13 × 21 gridded) sources, and the value of emissions were random from 10 to 130 mol km$^{-2}$ h$^{-1}$. The mean value of EM_real was 50 mol km$^{-2}$ h$^{-1}$. The background emission (EM_back) also included 273 sources having the same spatial distribution as EM_real. The value of EM_back conformed to a Gaussian distribution with a mean value of 50 mol km$^{-2}$ h$^{-1}$ and a variance of 10 mol km$^{-2}$ h$^{-1}$. That is to say, a random perturbation, which conformed to a Gaussian distribution, was added to a mean value of 50 mol km$^{-2}$ h$^{-1}$. The correlation coefficient between EM_real and EM_back was 0.01, suggesting two emissions were unrelated.

**Commemt2b:** Regardless, I don't feel the material about the OSSE adds much to the manuscript, and it is little more than a "sanity check" that I didn't find very convincing. I think a better demonstration of the 4DVAR system's efficacy would be to show plots of: 1) The cost function reduction from a 4DVAR analysis that assimilated real observations; 2) Analysis increments directly showing how $SO_2$ emissions were modified by assimilating $SO_2$ observations; and 3) "observation minus background" and "observation minus analysis" statistics. I think presenting these types of plots would increase confidence that your 4DVAR DA system works as expected and can provide stronger evidence of system robustness than the OSSE.

**Response 2b:** Thank you for your suggestion. The OSSE was deleted in the revised manuscript. As your advice, the performance of the 4DVAR system at 0000 UTC 17 January 2019 has been added in the revised manuscript to estimate the 4DVAR system's efficacy that assimilated real observations.

Figure R1 shows the result of Emi_2019 (Table 2 in manuscript) for the first day of 17 January 2019 to test the effect of the 4DVAR. The experiment employed MEIC_2016 as the background emission, and assimilated the hourly surface $SO_2$ observations during 0000–0600 UTC. In Fig. R1a, the observed heavy polluted areas with $SO_2$ concentrations exceeding 80 μg m$^{-3}$ are most located in North China Plain and Northeast China, and the observed light polluted areas with $SO_2$ concentrations

below 40μg m$^{-3}$ are most located in Southern China. Compared with the observed concentration, the background concentrations (Fig. R1b) underestimated in North China Plain, Northeast China, but overestimated in Central China and Sichuan Basin. Figure R1c shows the increment field of SO$_2$ concentrations that is the difference of analyzed field minus background field. There are positive values in most of Northern China and negative values in Central China and Sichuan Basin, suggesting that the optimized IC is more consistent with the observed SO$_2$ concentrations. The evaluation of the optimized IC and background concentrations are shown in Fig. R1d. Compared with the background field, the bias in analysis field improved from -2.8 to 1.8 μg m$^{-3}$, the RMSE decreased from 23.1 to 11.8 μg m$^{-3}$ and the correlation coefficient (CORR) of analysis field increased from 0.2 to 0.8. The result indicates that there is an improvement in the accuracy of the SO$_2$ concentration of IC by using 4DVAR method. The forecast accuracy with optimized IC can be improved (Peng et al., 2017, 2018), but the most important influencing factor for forecast accuracy is the emissions. The emissions and concentration IC can be optimized simultaneously by EMI_2019 experiment using our 4DVAR system.

[Figure]

(a) Observations        (b) background concentrations

(c) the increment of SO$_2$ concentrations       (d) scatter plots

[Figure]

**Figure R1: The simulated and observed SO₂ concentrations at 0000 UTC 17 January 2019. (a) Observations, (b) background concentrations, (c) the increment of SO₂ concentrations, and (d) scatter plots. Units: μg m⁻³.**

Figure R2a presents the background emission of MEIC 2016 at 0000 UTC. The heavy emission is found in North China Plain and Central China, Yangtze River Delta and Pearl River Delta. The largest emission values in these areas exceed 80 mol km⁻² h⁻¹. But the emissions in Northeast China are relatively low and generally less than 40 mol km⁻² h⁻¹. According to Fig. R1a and R1b, MEIC_2016 underestimated in most of Northern China and overestimated in Central China and Sichuan Basin. Fig. R2b shows the increment of SO₂ emissions at 00UTC 17 January 2019 by using the 4DVAR system. There are positive increment in North China Plain, Northeast China and negative increment in Central China and Sichuan Basin. Obviously, the distribution of the increment of SO₂ emissions is consistent with that of the increment of SO₂ concentration (Fig. R1c). There is a reasonable relationship between the two increments, since the underestimated/overestimated emission may result in underestimated/overestimated concentration for the simulation of SO₂.

(a) background emission                    (b) The increment of emission

[Figure]

[Figure]

**Figure R2: (a) The SO$_2$ background emissions and (b) SO$_2$ emissions increment at 0000 UTC 17 January 2019. Units: mol km$^{-2}$ h$^{-1}$.**

Figure R3 shows the size of the cost function for each inner iteration during the DA process of 0000–0600 UTC 17 January 2019. In this example, the maximum number of iterations was ten considering the balance between calculation time and result accuracy. It shows that the cost function quickly converges with an increase in the number of iterations. After eight iterations, the cost function was stable and close to minimum. The $J$ at the end iteration was 12% at the first iteration.

[Figure]

**Figure R3: Cost function for each inner iteration during the DA process of 0000–0600 UTC 17 January 2019.**

This statement has been added in the revised manuscript.

**Commemt2c:** Additionally, I'm somewhat concerned with Fig. 5b, which shows that the cost function increased between the 8th and 9th iterations. Variational cost

functions must monotonically decrease with each inner-loop iteration, so there is a chance that something went wrong. Please look into this or offer an explanation.

**Response 2c:** In our 4DVAR system, there are inner and outer loops. In an outer loop, there are ten (in our system) inner loops. During the run of each outer loop, ten cost functions are outputted to represent the minimum in a lower spatial resolution. For the different outer loop, the nonlinear trajectories and innovations are updated at high resolution, and the cost function are also updated. And the cost function after outer loop monotonically decrease. Note the outer loop is to provide some nonlinear information back into the minimization scheme, especially if the increment has move towards the limit of the viability of the tangent linear approximation about the current nonlinear trajectory.

The cost function in Fig. 5b (in the origin manuscript) was an attempt of the cost function in the inner loops, not the last cost function in this step (outer loop). The output of the real cost function was corrected in the 4DVAR system. Fig R4 shows the cost function of the OSSE. The cost functions decreased with each inner-loop iteration. After eight outer loop iterations, the cost function was stable and close to minimum.

[Figure]

**Figure R4: Cost function for each inner iteration in the OSSE.**

**Comment3a:** I think section 2.4 needs to be more specific about what you actually did, rather than making fairly general statements. What specific observation error did you use? What were your values of $\varepsilon_r$ and $\varepsilon_o$ and how did you arrive at these values?

**Response 3a:** The observation errors include the measurement error and the representative error. The observation error of SO₂ concentration $\varepsilon_{SO_2}$ is defined as below:

$$\varepsilon_{SO_2} = \sqrt{\varepsilon_r{}^2 + \varepsilon_o{}^2} \tag{2}$$

where $\varepsilon_o$ is the measurement error, and $\varepsilon_r$ is the representative error. The measurement error $\varepsilon_o$ is the systematic error generated during monitoring by the instrument at each environmental monitoring station. The representative error $\varepsilon_r$ represents the weight of observed data in the data assimilation system. Thus, the measurement error $\varepsilon_o$ of SO₂ observation in this study is defined as $\varepsilon_o = 1.0$ µg m⁻³ following the result of Chen et al. (2019).

The representative error $\varepsilon_r$ is caused by converting the model variable to the observation variable (Schwart er al., 2012) and can be expressed as:

$$\varepsilon_r = \gamma \varepsilon_o \sqrt{\frac{dx}{L}} \tag{3}$$

where $\gamma$ is an adjustable parameter scaling $\varepsilon_o$. $\gamma = 0.5$ was used, which is same as Dai et al., (2021). $dx$ is the grid spacing (27km in this study) and $L$ is the radius of influence of an observation, which was taken to be 2km following Chen et al. (2019). Then, $\varepsilon_r = 1.8$ µg m⁻³ calculated from Eq. (3).

This statement had been added in the revised manuscript.

**Comment3b:** Similarly, in lines 201-204, please state how you produced these 48- and 24-h forecasts. What model configuration did you use? You should also cite the "NMC method" (Parrish and Derber 1992) for this approach. Moreover, in line 202, should "state variables" be "background errors"? How did you model the correlations in $C$, especially for the emissions? Overall, please be more specific about your background and observation error covariance construction.

**Response 3b:** Thank you for your suggestion. In Eq. (1), $\mathbf{B}_{e_i}$ is the background error covariance (BEC) of SO₂ emission that was estimated from the background emission. $\mathbf{B}_c$ is the background error covariance (BEC) of SO₂ concentration that was estimated

by national meteorological center (NMC) method. The details of estimate $\mathbf{B}_{e_i}$ and $\mathbf{B}_c$ had been added in the revised manuscript as followed.

The BEC is too large to be handled numerically, we thus followed the method used by Li et al. (2013) and Zang et al. (2016) to simplify $\mathbf{B}$

$$\mathbf{B} = \mathbf{DCD}^{\mathrm{T}} \tag{4}$$

where $\mathbf{D}$ is the RMSE matrix and $\mathbf{C}$ is the correlation matrix.

$\mathbf{C}$ can be simplified by the Cholesky factorization and Kronecker product method (Li et al., 2013) as:

$$\mathbf{C}^{\frac{1}{2}} = \mathbf{C}_x^{\frac{1}{2}} \otimes \mathbf{C}_y^{\frac{1}{2}} \otimes \mathbf{C}_z^{\frac{1}{2}} \tag{5}$$

The NMC method (Parrish and Derber, 1992) was used to estimate the BEC of $SO_2$ concentrations. The differences between 48 h and 24 h forecasts were generated from 17 January 2020 to 18 February 2020. The first initial chemical field at 0000 UTC on 17 January 2020 was obtained from a 10 d forecast to clear away the effect of spin-up. The subsequent initial chemical fields were derived from the former forecast one day prior. The horizontal length scale was used to determine the magnitude of $SO_2$ variance in the horizontal direction. This scale can be estimated by the curve of the horizontal correlation with distances, and the horizontal correlation is approximately expressed by a Gaussian function $\mathbf{e}^{\frac{(x1-x)^2}{2L_s^2}}$ ($\mathbf{e}$ is the base of natural logarithms equal to 0.272). Here, $x1$ and $x$ are two points, and $L_s$ is the horizontal length scale. According to Zang et al. (2016), when the intersection of the decline curve reaches $\mathbf{e}^{1/2}$, the distance can be approximately as the horizontal length scale in Fig R5(a). The horizontal length scale was 81 km, which is approximately three-times larger than the scale used in this study. The vertical variance of $SO_2$ concentrations was considered by vertical correlations in the BEC. A strong relationship was observed in the boundary layer (approximately below the 20th model layer) in the vertical direction (Fig. R5(b)). The standard deviation demonstrates the reliability of the forecasting model, and the standard deviation for the vertical distribution of $SO_2$ concentrations decreased with increasing height in the $\mathbf{B}_c$ (Fig. R5(c)).

[Figure]

**Figure R5.** The background error covariation of SO₂ concentrations. (**a**) Vertical distribution of the horizontal correlation; the horizontal thin black line is the reference line ($e^{1/2}$) for determining the horizontal correlation scales. (**b**) Vertical correlations. (**c**) Vertical distribution of the standard deviation.

For the $\mathbf{B}_{e_i}$, the standard deviation $\mathbf{D}_{e_i}$ is diagonal with a 200% error (Wang et al., 2012) and $\mathbf{C}_{e_i}$ is a block diagonal matrix, with the main diagonal blocks being the correlation matrices of SO₂ emission. The main diagonal blocks of $\mathbf{C}_{e_i}$ is 1.0 because the emission in each grid point is independent with other grids.

**Comment4a:** Several aspects of the experimental design were not initially clear to me and caused confusion. Although some elements became clearer with time, I think descriptions of the experiments should be clarified:

a. From Table 3, it appears that you performed DA experiments for ~3 weeks. What was the cycling period of your experiments (i.e., how often did you produce new 4DVAR analyses)? Was it 6 hours? Did you continuously cycle both the chemistry and meteorology, or did you periodically update meteorology from an external source, like the GFS model? What did you do for chemical boundary conditions (my apologies if I missed it)? Furthermore, Table 3 states 24-h forecasts were produced, but how often did you initialize these 24-h forecasts? Overall, the temporal aspect of the experiments should be clarified.

**Response 4a:** Thank you for your insightful and constructive question. Yes, the 4DVAR cycling period is 6 hours. We performed two sets of DA experiments to obtain optimized emissions of 2019 and 2020 (Table R1). For the set of experiments of Emi_2019, the first DA process started on 17 January 2019, and the observations during 0000–0600 UTC were assimilated by Eq. (1). The MEIC_2016 0000–0500 UTC emissions were the background emissions. The assimilated SO₂ concentration

initial field (0000 UTC) and the optimized SO$_2$ emission during 0000–0500 UTC were obtained.

Table R1 shows the details of DA emissions experiments. For the set of experiments of Emi_2019, the first DA process started on 17 January 2019, and the observations during 0000–0600 UTC of 17 January 2019 were assimilated by the 4DVAR system. Then, the optimized SO$_2$ concentration initial field (0000 UTC) and the optimized SO$_2$ emission during 0000–0500 UTC were obtained. Before conducting Emi_2019 experiment, 24 h forecasts were performed by WRF-Chem with MEIC_2016 emissions every 0000 UTC from 17 January to 7 February, 2019 to provide physical and chemical parameter. The chemical ICs of each day were obtained from the 24 h forecasting of the previous day. For the 24 h forecast, the meteorological initial and boundary conditions were provided by the 1° × 1° National Centers for Environmental Prediction (NCEP) Global Final Analysis data at a 6-hour frequency. The chemical boundary fields were not considered because the domain used in this study was wider than China. For the experiment of Emi_2019, the emission of 2019 were optimized by 4DVAR system every 6 hours with the background emissions of MEIC_2016. The physical and chemical parameter used in this DA process were obtained by the WRF-Chem forecast. For the experiment of Emi_2020, the DA process settings are similar with the Emi_2019 experiment. The optimized emissions of 2020 is obtained with the emission 2019 as background emission.

**Table R1: Details of 4DVAR experiments to optimize the emission for 2019 and 2020.**

| Name | Background emissions | Optimized emissions | Study period |
|---|---|---|---|
| Emi_2019 | MEIC_2016 | 2019 optimized emissions | Every 6 hours from17 January to 7 February, 2019 |
| Emi_2020 | 2019 optimized emissions | 2020 optimized emissions | Every 6 hours from 17 January to 7 February, 2020 |

To estimate the improvement of SO$_2$ forecasts using optimized emissions, three sets of forecast experiments were performed using the MEIC_2016 emissions and the optimized emissions for 2019 and 2020, respectively, and these were labeled

Ctr_2016, DA_2019, and DA_2020, respectively (see Table 3).. The three experiments were run daily with 24 h forecasts from 17 January to 7 February 2020. All experiments used the same WRF-Chem domain settings and physiochemical parameters. The $SO_2$ initial condition (IC) at 0000 UTC on January 17 was based on the spin-up forecasts initialized at 0000 UTC on January 7, 2020 for all three forecast experiments. The $SO_2$ ICs were later obtained from the 24h forecasting of the previous day for the three experiments, respectively. For example, the $SO_2$ IC of the experiment beginning at 0000 UTC on 18 January was from the 24h forecast result of the experiment beginning at 0000 UTC of 17 January, and so on. Meteorological initial and boundary conditions were provided by the 1° × 1° NCEP Global Final Analysis data at a 6-hour frequency. The chemical boundary fields were not considered.

**Table R2: Details of the forecast experiments with emissions of 2016, 2019 and 2020.**

| Name | Emission | Forecast duration | Study period |
|------|----------|-------------------|--------------|
| Ctrl_2016 | MEIC_2016 | 24 h | Every day from 17 January to 7 February, 2020 |
| DA_2019 | The 2019 optimized emissions | 24 h | Every day from 17 January to 7 February, 2020 |
| DA_2020 | The 2020 optimized emissions | 24 h | Every day from 17 January to 7 February, 2020 |

b.    Fig. 3 didn't seem clear to me. What specific field(s) are being updated?    Just $SO_2$ emissions? Or both $SO_2$ emissions and concentrations? Additionally, this figure might be clearer if you annotated the mathematical symbols from Eq. (1) or (3) on it so readers can link this figure to the equations. It might also be nice if you added another panel to the figure showing the temporal progression of the DA system (per above comment). Finally, in the top-left box, there's a typo (it should be "field").

**Response 4b:** Thank you for your suggestion. Figure 3 in the original manuscript has been modified. In the 4DVAR system, both $SO_2$ concentration initial field ($c_0$) and $SO_2$ emissions ($e_i$) were updated as the state vector. The mathematical symbols from Eq. (1) had been added in Fig. R6.

[Figure]

Figure R6: Flow chart of the SO₂ emissions optimization procedure in a single time step of $i$. The orange boxes represent the SO₂ optimized emissions and SO₂ concentrations of output. The $c_0^b$, $c_0$, $e_i^b$ $e_i$ and $y_i^o$ are the mathematical symbols from Eq. (1).

    c. It appears that you ran the 4DVAR DA system over two separate periods to estimate SO₂ emissions: 1) 17 Jan – 7 Feb 2019; and 2) 17 Jan – 7 Feb 2020. However, you never explicitly stated this! Thus, there are really two parts to this work. The first is estimating emissions in 2019 and 2020 from the 4DVAR DA system. The second is using different emissions estimates to drive various sets of forecasts over a common period in 2020. This distinction was not always clear, which caused me confusion.

    Please explicitly state these various experiments and their purposes. It is important that you do so because of places like line 328, where you simply stated the *years* of the emissions, and not the names in Table 3 (which contain years); more distinction needs to be made for data *for a given year* versus various experiments *in 2020* that used emissions generated from various years.

**Response 4c:** Thank you for your suggestion. In this study, we performed two sets of DA experiments to obtain optimized emissions of 2019 and 2020 (Table R1) and three sets of forecasting experiments (Table R2) to estimate the improvement of SO₂ forecasts using optimized emissions. The statement of the experiments has been modified in the revised manuscript. Please see our **Response 4a** to previous comment

4a.

> d. For the DA_2019 experiment, you effectively seemed to use a "pre-processing" step, where you ran the 4DVAR DA system over 2019 and then used those emissions when simulating a period in 2020. It might be worth noting that in contrast, the DA_2020 experiment did everything all at once without the need for a "pre-processing" step.

**Response 4d:** The "pre-processing" is to prepare background emission, chemical initial conditions, meteorological initial and boundary conditions. For the forecast experiments (Ctrl_2016, DA_2019 and DA_2020), all settings were the same in three forecast experiments, except for the emissions.

**Comment5:** Section 3.2: Please clarify that the data (i.e., emissions) discussed in this subsection were obtained from the 4DVAR analyses (I think), and not some other source. In general, please be more precise about from where the data on each figure come from.

**Response 5:** Thank you for your suggestion. The hourly $SO_2$ concentrations were taken from the China National Environmental Monitoring Center (http://www.cnemc.cn), and the 2016 $SO_2$ emission were obtained from the MEIC (http://www.meicmodel.org/). The 2019 and 2020 $SO_2$ emissions were obtained from the 4DVAR analyses.

The statement has been added in the Section 2.4 in the revised manuscript.

**Comment6:** The writing is understandable but there are many grammatical errors that I found distracting. Please carefully proofread the manuscript.

**Response 6:** The grammatical errors were corrected in the revised manuscript. Below are our point-to-point response in detail.

**Minor comments**

1. Line 48: "most explored algorithms" for what? Please be specific.

Corrected.

2. Lines 50-51: The second instance of "to estimate" should be removed, and perhaps "total regional and global emissions" should be moved to after the first instance of "to estimate".

Corrected.

3. Lines 60-63: This statement is too broad. There are techniques to handle this problem, like inflation, that are well established at least for meteorological EnKF DA. Please refine this statement.

Accepted. The statement had been deleted, and the previous EnKF $SO_2$ estimates had been added in the revised manuscript.

4. Line 89: Please add a reference for WRF-Chem.

Corrected.

5. Line 105: Suggest "covering all of China" instead of "covering the entire country".

Corrected.

6. Lines 105, 158, 159: "resolution" should be "grid spacing", as the two are not the same.

Corrected.

7. Lines 111, Table 1: Please add a reference for the Grell-3D scheme.

Corrected.

8. Fig. 1 caption: Please state that this figure also shows the WRF modeling domain!

Corrected.

9. Lines 122-125: Somewhere in here, please specifically define $n$.

Corrected.

10. Somewhere in section 2.2, please be more precise about which "control variables" are included in $c_o$. Is it just $SO_2$ concentration?

Corrected. $c_0$ and $e_i$ are the control variable that denotes the initial concentration vector and the modified emission.

11. Line 131: Here, is $H$ nonlinear? (probably it is). Please state.

*H* is linearization, which operates on the model grid ($SO_2$ simulated concentrations) to generate a best estimate of the observed value ($SO_2$ concentrations). The statement of *H* has been revised.

12. Lines 131-132: Please state that $R$ is the observation error covariance matrix.

Corrected.

13. Line 133: Please define $f$.

$f_{i,i-1}$ represents the model time integration for one time step from time $i - 1$ to $i$. The statement had been added.

14. Eq. (3): Some more explanation is needed about how you go from Eq. (1) to Eq. (3) for the observation term. Specifically, please note the linearization about the background.

Thank you for your suggestion. Eq. (1) to Eq. (3) are only converted the equations from the solved of objective function to the incremental form. *H* plays a role of interpolation from the model grid to the observed value. Thus, *H* is linear. The tangent linear operators $\Gamma$, $L$ (in Eq. (4-7) are derived from WRF-Chem are very complex and computational demanding, thus, we simplify the CTM to focus on $SO_2$.

Equation (8) in the manuscript is the governing equation for the concentrations:

$$\frac{\partial c}{\partial t} = - u\frac{\partial c}{\partial x} - v\frac{\partial c}{\partial y} - w\frac{\partial c}{\partial z} + \frac{\partial}{\partial x}\left(K_x \frac{\partial c}{\partial x}\right) + \frac{\partial}{\partial y}\left(K_y \frac{\partial c}{\partial y}\right) + \frac{\partial}{\partial z}\left(K_z \frac{\partial c}{\partial z}\right) - \mathbf{e}^{-\Lambda}\frac{\partial c}{\partial t} -$$

$$r\frac{\partial c}{\partial t} + V_m \frac{\rho_{air}}{\rho}\frac{\Delta S}{dz}e \qquad (8)$$

In Eq. (8), the changes in concentrations are linear and only relate to the physicochemical parameters, such as $u, v, w, K_z, \Lambda, r, \rho$, and $e$.

We used the values ($u, v, w, K_z, \Lambda, r, \rho$, and $e$) within an integration step (10mins) to represent the mean of these variables in the 4DVAR system. This process would lead some errors due to the linear operators. But even we used a shorter integration step of 2mins, the result is close to that of the integration step of 10mins (Fig. R7). The average difference in concentrations between the two experiments was 0.3 in the grid of $i =$

$94$ , $j = 152$ during 1 hour (Fig. R7a), which was also 1% of the total average concentrations. The mean difference in concentrations over China was 0.1 (Fig.R7b). Thus, it is concluded that the error from the linearization is very small and negligible.

[Figure]

Figure R7: The forecast concentrations in the forward process by using different values in: (a) the grid of $i = 94$ , $j = 152$ and (b) China.

15. Lines 150-152: How does this relate to the cycling period of the DA system? Does this mean you produced new analyses every 6?

Yes, the 4DVAR system was performed every 6 hours to obtain the optimized emission. For example, the first DA process started at 17 January 2019, and the observations during 0000–0600 UTC were assimilated by Eq. (1). The MEIC_2016 0000–0500 UTC emissions were the background emissions. The assimilated $SO_2$ concentration initial field (0000 UTC) and the optimized $SO_2$ emission during 0000–0500 UTC were obtained.

16. Line 161: "large horizontal resolution study" is unclear. Are you referring to your specific study or something else? Please clarify.

Sorry for the misleading. The horizontal resolution is 27km in this study. Thus, the $\frac{\partial}{\partial x}(K_x \frac{\partial c}{\partial x}) + \frac{\partial}{\partial y}\left(K_y \frac{\partial c}{\partial y}\right)$ can be neglect. The statement has been corrected.

17. Eq. (10): $L_{turb}$ doesn't appear in the list of quantities in line 169, and $L_{dry}$, which does appear in line 169, doesn't appear in the list of equations.

Please clarify. Also please double check Eq. (14).

It should be $L_{turb}$ in line 169. The statement had been deleted in the revised manuscript.

18. Lines 170-185: Should $L_{trub}$ be $L_{turb}$?

    Corrected.

19. Line 180: Should it be Eqs. 9–13 instead of Eqs. 9–12?

    Corrected.

20. Lines 168-185: I'm not an expert about adjoint modeling, but I had the feeling that these lines aren't precise enough about the adjoint model formulation. Shouldn't there be more derivatives in there?

    In our 4DVAR system, the adjoint was applied to calculate $\Gamma^T$, $L^T$ and $\boldsymbol{H}^T$ in Eq. (6) and (7), which are derived from the tangent linear model operator $\Gamma$, $L$, and observation operator $\boldsymbol{H}$. $\boldsymbol{H}$ plays a role of interpolation from the model grid to the observed value and is linear, thus $\boldsymbol{H}^T$ is easily derived using tangent linear coding techniques. The tangent linear operators $\Gamma$, $L$ are simplified from the WRF-Chem model (Eq. (8)).

    Using tangent linear coding techniques, we derived the code for the discretized tangent linear operators $\Gamma$, $L$ from the source code built in WRF-Chem. Once the source code is available for the tangent linear operators, we use the adjoint coding technique to derive the adjoint operator. The adjoint coding technique are detailed in Hoffman et al. (1992).

    This statement has been added in the revised manuscript in section 2.2.

21. Line 192: Please omit "the assimilation variable, which is the"…it's confusing, because that phrase is somewhat referring to a state/control variable, even though you're really talking about observation errors.

    Corrected

22. In lines 220-228, please be very precise about "emission" vs. "concentration" in your descriptions.

    Corrected.

23. Line 243: Please change February 6 to February 7 for consistency with Table

3.

Corrected.

24. Line 246: Typo: it should be "physiochemical".

Corrected.

25. Line 249: Suggest "…based on the spin-up forecasts initialized at 0000 UTC…".

Corrected.

26. Line 250: Please be more precise about "the previous day" (this comment relates to earlier comments about the cycling period).

Corrected.

27. Line 256: There seem to be more than 13 x 9 points in Fig. 4, so I was confused about this statement concerning "arrays and columns". Please clarify.

Sorry for misleading, it is 13 x 21 points.

28. Fig. 5: The legend in the left panel is covering data and should be moved, and the y-axis in the right panel should probably be "J" not "$J_b$".

Corrected.

29. Line 282: I'm not sure I agree with this statement, especially in (a) and (b); the 2019 $SO_2$ concentration decreases with time but the $SO_2$ emissions seem steady. Please revise.

Corrected.

30. Line 283: To my eyes, it looks like the lowest emissions were on 1 February, not 3 February (per Fig. 6b).

Corrected.

31. Lines 289-292: Please point to Figs. 6c,d here.

Corrected.

32. Fig. 6 caption: Please state the meanings of the vertical lines.

Corrected.

33. Line 311: I believes "rates" should be "ratios".

Corrected.

34. Fig. 7: What are the insets in the lower right corner of each panel? Additionally, please be more precise about the subtraction convention. Above (c) and (d), it says "2020–2019" but the caption says, "differences between 2019 and 2020", which implies "2019–2020". It might be clearest to just write out "2020 minus 2019". Finally, please state in the caption whether these statistics are averaged over the entire period. Similar comments also apply to Fig. 8.

The inset in the lower right corner of each panel is South China Sea, which belongs to China. Done as suggested.

35. Line 315: I don't think "observations" is the correct word. Is "analyses" more accurate? Please also see line 363.

Corrected.

36. Line 320: Please remove "slightly"; it's too subjective.

Corrected.

37. Line 321: Please remove "Remarkably", which is also subjective, and furthermore, the differences don't seem "remarkable".

Corrected.

38. Line 333: Please remove "slightly". Also, it seems that this behavior was only evident in Fig. 9a, so please clarify the region you are discussing.

Corrected.

39. Fig. 9 caption: Are these statistics averaged/aggregated over the entire period and over all sites or grid points? Please clarify.

Yes, the statistics are averaged over the entire period and over all grid points. The statement has been revised.

40. Lines 352-355: I found this chunk troublesome. The explanation you offered didn't make sense to me, and I'm not sure all your statements are accurate. Please clarify or omit.

Thank you for your suggestion. The explanation has been deleted.

41. Throughout, including figure captions: "Skill" should be "accuracy". Skill

is "accuracy relative to a baseline", and all of the metrics you are showing are measures of accuracy, not skill. I believe every instance of "skill" needs to be changed to "accuracy".

Corrected.

42. Fig. 10 caption: Please clarify whether these statistics are averaged/aggregated over the entire time period and all sites. Same comment for Fig. 11.

The statistics are averaged over all sites in China. Corrected.

43. Lines 369-370: Please omit "compared with the Ctrl_2016 experiment".

Corrected.

44. Lines 373-374: Please clarify what you mean by the "background field". Do you mean the field at the every start of the period (0000 UTC 17 January 2020)?

No, the background field means the background emission. For Emi_2019 and Emi_2020 experiments (Table R1), the 4DVAR cycling period is 6 hours. For Ctrl_2016, DA_2019 and DA_2020 experiments (Table R2), 24 h forecasts were performed every 0000 UTC from 17 January 2020 to 7 February 2020. The statement was revised.

45. Line 390: Can you point to a figure for this key result about the decrease of optimized emissions? Also, did you ever state these values in the results section (sorry if I missed it)?

The mean optimized emission and increment $SO_2$ emission field was shown in Fig. in the manuscript. And as your advice, the performance of the 4DVAR system at 0000 UTC 17 January 2019 had been added in the revised manuscript. Please see our **Response 2b** to previous major comment 2b.

46. Figs. 1, 4, 5, 8, 9: Please add annotations (e.g., "a", "b") to all these figures.

Corrected.

**Reference**

Chen, D., Liu, Z., Ban, J., Zhao, P., and Chen, M.: Retrospective analysis of 2015–2017 wintertime PM2.5 in China: response to emission regulations and the role of meteorology, Atmos. Chem. Phys., 19, 7409-7427, 10.5194/acp-19-7409-2019, 2019.

Dai, T., Cheng, Y., Goto, D., Li, Y., Tang, X., Shi, G., and Nakajima, T.: Revealing the sulfur dioxide emission reductions in China by assimilating surface observations in WRF-Chem, Atmos. Chem. Phys., 21, 4357-4379, 10.5194/acp-21-4357-2021, 2021.

Henze, D. K., Hakami, A., and Seinfeld, J. H.: Development of the adjoint of GEOS-Chem, Atmos. Chem. Phys., 7, 2413-2433, 10.5194/acp-7-2413-2007, 2007.

Hoffman, R., Louis, J. F., and Nehrkorn, T.: A method for implementing adjoint calculations in the discrete case, in, ECMWF, Shinfield Park, Reading, https://www.ecmwf.int/node/9906, 1992.

Li, Z., Zang, Z., Li, Q. B., Chao, Y., Chen, D., Ye, Z., Liu, Y., and Liou, K. N.: A three-dimensional variational data assimilation system for multiple aerosol species with WRF/Chem and an application to PM2.5; prediction, Atmospheric Chemistry and Physics, 13, 4265-4278, 10.5194/acp-13-4265-2013, 2013.

Parrish, D. F., and Derber, J. C.: The National Meteorological Center's Spectral Statistical-Interpolation Analysis System, Monthly Weather Review, 120, 1747-1763, 10.1175/1520-0493(1992)120<1747:TNMCSS>2.0.CO;2, 1992.

Schwartz, C. S., Liu, Z., Lin, H.-C., and McKeen, S. A.: Simultaneous three-dimensional variational assimilation of surface fine particulate matter and MODIS aerosol optical depth, Journal of Geophysical Research: Atmospheres, 117, https://doi.org/10.1029/2011JD017383, 2012.

Zang, Z., Li, Z., Pan, X., Hao, Z., and You, W.: Aerosol data assimilation and forecasting experiments using aircraft and surface observations during CalNex, Tellus B: Chemical and Physical Meteorology, 68, 10.3402/tellusb.v68.29812, 2016.

Zheng, B., Tong, D., Li, M., Liu, F., Hong, C., Geng, G., Li, H., Li, X., Peng, L., Qi, J., Yan, L., Zhang, Y., Zhao, H., Zheng, Y., He, K., and Zhang, Q.: Trends in China's anthropogenic emissions since 2010 as the consequence of clean air actions, Atmospheric Chemistry and Physics, 18, 14095-14111, 10.5194/acp-18-14095-2018, 2018.

---

## Author Comment (AC3)

**Response to the Comments of Referees**

**Manuscript ID: acp-2022-301**

**Title:** Four-dimensional Variational Assimilation for SO2 Emission and its Application around the COVID-19 lockdown in the spring 2020 over China

Author: Yiwen Hu, Zengliang Zang\*, Xiaoyan Ma\*, Yi Li, Yanfei Liang, Wei You, Xiaobin Pan, Zhijin Li

We thank the reviewers and editors for providing helpful comments to improve the manuscript. We have revised the manuscript according to the comments and suggestions of the referees.

The referee's comments are reproduced (black) along with our replies (blue). All the authors have read the revised manuscript and agreed with submission in its revised form.

**< Anonymous Referee #3>**

**Comment:** A timely and accurate emission is important for atmospheric chemistry simulation and pollution control. It is challenging and difficult to estimate the emission by using the "top-down" approach of 4DVAR. To my knowledge this is the first time when the 4DVAR system is development for optimizing SO2 emission and applied to investigate SO2 emission changes during the COVID-19 lockdown. The results shows that there is a significant decrease of SO2 emission between 2019 and 2020 due to the COVID-19 lockdown. It is reasonable and helpful for the improvement of atmospheric chemistry forecast. I suggest publishing this paper after the following points are addressed.

**Response:** We thank the referee for the positive comments on our manuscript. The manuscript has been carefully revised according to the referee's comments and suggestions.

**Comment 1:** In the introduction, I suggest the author add some descriptions of emission optimization with the EnKF method.

**Response 1:** Thanks for your suggestion. For the EnKF method, many studies estimated SO2 emissions by assimilating surface and satellite observational data in recent years, such as Dai et al (2021), Chen et al. (2019), Koukouli et al. (2018) and so on. Dai et al. (2021) developed a four-dimensional regional ensemble transform Kalman filter and showed that the SO2 emissions over China in November 2016 decreased 49.4% in comparison to the 2010 background emission due to the implementation of emission control policies (Zheng et al., 2018).

Above literature review has been added in the introduction and discussion of revised manuscript.

**Comment 2:** In Fig. 2, how does the author classify the assimilating stations and verifying station?

**Response 2:** There are 1933 national control measurement stations in China in January 2020. The stations were gridded into the model grid  $(27 \times 27 \ km^2)$ . If there were more than 2 stations located in the same grid, one station was randomly selected to verify the improvement in using optimized emissions, and the remaining stations were used for assimilation. In this study, 508 stations were selected for verifying, while the remaining 1425 stations were used to assimilate.

This statement has been added in the revised manuscript.

**Comment 3:** In Fig. 3, there is not a box of observation in the flow chart. In addition, the variable of output is only SO2 emission. It should be added the initial SO2 concentration, since both the  $SO_2$  concentration and the emission are the state variables in this study.

**Response 3:** Thanks for your suggestion. Figure 3 has been revised in the manuscript. The box of SO2 observation (input) and SO2 concentration (output) has been added.

Figure R1: Flow chart of the SO2 emissions optimization procedure in a single time step of *i*. The orange boxes represent the SO2 optimized emissions and SO2 concentrations of output. The  $c_0^b$ ,  $c_0$ ,  $e_i^b$   $e_i$  and  $y_i^o$  are the mathematical symbols from Eq. (1).

**Comment 4:** Why the author firstly optimized the SO2 emission of 2019 from the emission of 2016. Did the author directly optimize the emission of 2020 from the emission of 2016?

**Response 4:** The goal of this study is to optimize the SO2 emissions in January 2020 by the 4DVAR method and evaluate the influence of COVID-19 on SO2 emissions. And the difference between 2019 and 2020 emissions during the same period reflected the influence of COVID-19 lockdown. However, since there were no 2019 emissions, we only first generated 2019 optimized emissions using the 4DVAR system, and the MEIC\_2016 was set as background emission. Then the 2020 emissions were optimized, and 2019 optimized emissions were set as background emissions.

**Comment 5:** The author did not show the increment field of  $SO_2$  concentration. I suggest the author add it. The scatter point of  $SO_2$  concentration between observation and assimilation also should be illustrated.

**Response 5:** Thanks for your suggestion. The increment of SO2 concentration and emissions at 0000 UTC 17 January 2019 have been added in the revised manuscript to estimate the 4DVAR system's efficacy that assimilated real observations.

Figure R2 shows the model simulated and observed SO2 concentrations at 0000 UTC 17 January 2019. The MEIC 2016 was the background emission, and the hourly surface SO2 observations during 0000–0600 UTC was assimilated. The observed SO2 concentrations (Fig. R2(a)) showed that the most polluted area was located in North China Plain and Northeast China, and the observed SO2 concentrations in southern China were generally lower than 20  $\mu$ g m-3. Compared with the observations, the SO2 background concentrations were overestimated in Southern China, especially in Central China, Sichuan Basin, and Pearl River Delta. In addition, the SO2 concentrations were underestimated in Northern China, Western China, and Southeast China. The increment of SO2 concentrations showed the same change trends with the difference of observation and background field (Fig. R2(c)), reflecting the improvement of SO2 concentration analysis field. Compared with the background field, the mean bias in analysis field improved from -2.8 to  $1.8 \ \mu g \ m^{-3}$ , and the RMSE decreased from 23.1 to 11.8 µg m-3. The CORR of analysis field increased from 0.2 to 0.8, suggesting the accuracy of SO2 analysis field were improved using 4DVAR method.

Figure R2(e) is the background emission from MEIC 2016 at 0000 UTC and Fig. R2(f) shows the increment of SO2 emissions at 0000 UTC 17 January 2019. The SO2 background emission (Fig. R2(e)) showed that there were high emissions in South China, especially in Central China, Sichuan Basin and Pearl River Delta. The increment of SO2 emissions (Fig. R2(f)) decreased the emissions in these regions. Zheng et al (2018) also found the emissions decreased in Southern China due to the implementation of emission reduction policies in China. The positive increment of SO2 emissions (Fig. R2(f)) was lower than 1 mol km-2 h-1 in Southeast China, but the increment of SO2 concentrations (Fig. R2(c)) was generally more than 10  $\mu$ g m-3, indicating the difference between background field and observation in Southeast

China was caused by the uncertainty of initial concentration field, not the emissions.

---

## Referee Report (RR1)

Second review of

**Four-dimensional Variational Assimilation for SO₂ Emission and its Application around the COVID-19 lockdown in the spring 2020 over China**

by

Yiwen Hu et al.

**Overall comments**

I appreciate the authors' revisions to the manuscript and believe it is stronger. In particular, the data assimilation system and experiments are more clearly described. I think the manuscript is nearly ready for publication and only have minor comments for the authors to further consider.

At this point, I think the biggest issue is the grammar. There are many errors and typos, and I encourage the authors to fix as many as they can prior to publication.

**Minor comments**

1. Line 28: Perhaps I missed it, but I didn't see this information about exactly 200% and 300% increases of correlation coefficient in the main text. This information should be noted in the main text if you want to state it in the abstract.
2. Line 30: 238.7% is inconsistent with lines 426 and 457. Line 426 says the increase of correlation was 238.7%, but line 457 says it was a 201.3% increase. The values in lines 30, 426, and 457 should be identical.
3. Line 107: Please point specifically to Fig. 1a.
4. Line 120: Please abbreviate Energy Golden Triangle in the caption.
5. Fig. 1 caption (line 124): Please remove "which belongs to China" (sounds too political).
6. Line 171: Please remove "large".
7. Lines 192-193: Please remove the sentence beginning with "The representative error…". It's out of place, and it's an also an inaccurate statement.
8. Line 195: Typo in the reference ("er al."), and I didn't catch this in the reference list (could have missed it).
9. Line 200: Did you mean Eq. (12) instead of Eq. (10)?
10. Lines 291-292: The statement "From the observed…" is not a proper sentence. Please revise.
11. Lines 300-302: Please make sure all these specific values in the text exactly match those annotated on the figure. For instance, in line 300, 2.8 should be 2.76 and 1.8 should be 1.79.
12. Fig. 5d: Instead of "control" and "DA" in the legend, please call it "background" and "analysis".
13. Lines 321-322: This sentence can be removed, as it's the same as lines 306-308.
14. Lines 330, 443, and elsewhere: Because you did not perform formal statistical significance testing, please don't use the word "significant". A better word is "substantial".
15. Fig. 7b: Are the units percent? If so, please state. Same comment for Fig. 9b.

16. Lines 364, 379, 400, and 449: Instead of writing "2019" and "2020", please write Emi_2019 and Emi_2020 for consistency with Table 2 and clarity.
17. Fig. 10: It might be helpful to change the black dotted lines to red dotted lines to more easily see the correspondence with 2020.
18. Lines 438-441: Please omit. These are not key findings that need to be recapped.